# ARC-AGI Without Pretraining

## Abstract

Conventional wisdom in the age of LLMs dictates that solving IQ-test-like puzzles from the ARC-AGI-1 benchmark requires capabilities derived from massive pretraining. To counter this, we introduce *CompressARC*, a model without any pretraining that solves 20% of evaluation puzzles by minimizing the description length (MDL) of the target puzzle purely during inference time. The MDL endows CompressARC with extreme generalization abilities typically unheard of in deep learning. To our knowledge, CompressARC is the only deep learning method for ARC-AGI where training happens only on a fraction of one sample: the target inference puzzle itself, with the final solution information removed. Moreover, CompressARC does not train on the pre-provided ARC-AGI "training set". Under these extremely data-limited conditions, we do not ordinarily expect any puzzles to be solvable at all. Yet CompressARC still solves a diverse distribution of creative ARC-AGI puzzles, suggesting MDL to be an alternative, highly feasible way to produce intelligence, besides conventional massive pretraining.

## 1 Introduction

The ARC-AGI benchmark poses a uniquely challenging problem: to construct a system capable of solving novel, abstract reasoning puzzles using only a handful of examples. [1] These puzzles are intentionally designed to measure generalization, creativity, and pattern recognition, and have historically resisted solutions by even the most powerful pretrained large language models (LLMs). The most successful attempts have leaned heavily on massive datasets, fine-tuning, or test-time augmentation. [2, 3, 4] However, one possible approach towards artificial general intelligence (AGI) has remained surprisingly underexplored in practice: the principle of minimum description length (MDL). [5] Closely related to Kolmogorov complexity [6], MDL frames intelligence as the ability to compress information efficiently into a minimally sized program, that correctly outputs the original information when run. Despite its elegant theoretical connection to generalization and prediction, MDL has rarely been successfully implemented in deep learning as an alternative source of intelligence to pretrained LLMs. In this work, we directly investigate the power of compression by introducing *CompressARC*, a deep learning method that minimizes description length at inference time: it has no prior training at all—and yet it still achieves modest performance on ARC-AGI.

CompressARC tries to harness MDL by using deep learning, a combination of techniques plagued with incompatabilities and roadblocks. The main difficulty in using deep learning to minimize the description length is that the description is a discrete program, and cannot be differentiated. Moreover, the size of the program varies as we optimize over the program's code, running counter to gradient descent's requirement of a fixed number of training parameters. Together, these two difficulties make it nearly inconceivable to use gradient descent for searching the description space. As a result, past MDL-based attempts to solve ARC-AGI have focused on search in (at least partially) discrete program spaces. [7] The powerful expressive capacity of deep neural networks, requiring gradient

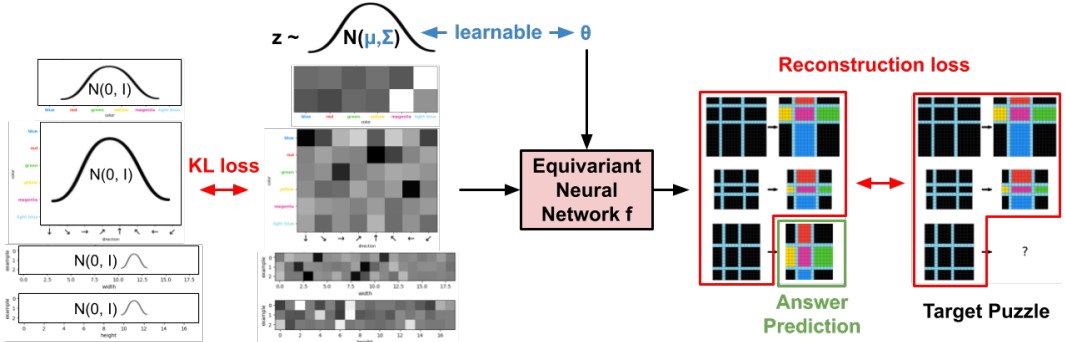

Figure 1: An overview of CompressARC, an MDL-derived deep learning solution to ARC-AGI. We learn some noise distribution $z \sim N(\mu, \Sigma)$ (left), feed it into a neural network, and compare the output to the puzzle we want to solve (right). We learn the noise and the weights at inference time to minimize the reconstruction error with the target puzzle, with a KL loss term that controls the noise distribution. The network's answer prediction is whatever it outputs for the last box (green).

descent to achieve, has not yet been fully combined with the strong generalization abilities promised by the MDL principle. These strengths are exactly what CompressARC has managed to conjoin.

The innovation that underlies CompressARC is a procedure for compiling the continuous information stored in a tensor into a discrete code. This procedure is special in that we can track the expected resulting code length from the perspective of the original continuous space, without ever having to perform the compilation, all in a differentiable fashion. This affords us the ability to include neural networks as part of the description, along with tensors representing their weights and inputs. The entire problem of minimizing the discrete description length is then offloaded as a deep learning task: the final procedure drawn in Figure 1. If we respect the restrictions imposed by the conversion of MDL into a deep learning problem, then we may enjoy MDL's strong generalization abilities as benefit:

- **No training time:** Since MDL requires us to start by having the target puzzle in hand, CompressARC starts by skipping training time, to go to inference time immediately to first obtain the target puzzle.

- **Inference time learning:** At this point, MDL dictates we minimize the description length, so CompressARC must run gradient descent using the target puzzle during inference time, to produce the solution.

- **Relaxed data requirement:** Since we expect to enjoy such strong generalization abilities endowed by MDL, we don't bother loading any other puzzles into memory. The target puzzle just by itself is already plenty of data.

Of course, this means CompressARC skips pretraining and leaves any training set puzzles unused. Even so, the extreme generalization of MDL allows CompressARC to solve 20% of evaluation puzzles and 34.75% of training puzzles, where we would ordinarily expect 0% from any traditional deep learning method under these conditions.

The remaining sections describe the ARC-AGI benchmark (Section 2), how CompressARC works (Section 3), CompressARC's architecture (Section 4), CompressARC's performance on ARC-AGI (Section 5), our interpretation of CompressARC's solution to an example puzzle (Section 6), and our conclusions (Section 7).

## 2   Background: The ARC-AGI Benchmark

ARC-AGI-1 is an artificial intelligence benchmark designed to test a system's ability to acquire new skills from minimal examples. Each puzzle in the benchmark consists of a different hidden rule, which the system must apply to an input colored grid to produce a ground truth target colored grid. Several input-output grid pairs are given as examples to help the system to infer the hidden rule

in a puzzle. The system is allowed **two attempts** to guess the output grid correctly, i.e., getting every single pixel color correct. The ARC Prize Foundation has launched competitions for machine solutions to ARC-AGI-1, with upwards of **$1,000,000** in prizes. [2, 8]

There are 400 training puzzles are easier than the 400 evaluation puzzles, and are meant to help your system learn the ideas of objectness, goal-directedness, numbers & counting, and basic geometry & topology. **These training puzzles play no role in the operation of CompressARC, and we only used them to inform our decisions of how to build CompressARC's architecture.**

The puzzles are designed so that **humans can reasonably find the answer, but machines should have more difficulty**. The average human can solve 76.2% of the training set, and a human expert can solve 98.5%. [9] Current methods for solving ARC-AGI focus primarily on tokenizing the puzzles and arranging them in a sequence to prompt an LLM for a solution, or code that computes a solution. [3] Top methods typically fine-tune on augmented training puzzles and larger alternative synthetic puzzle datasets [10] and test-time training [4, 11]. Reasoning models have managed to get up to 87.5% on the semi-private evaluation set, albeit with astronomical amounts of compute. [12]

Please refer to Appendix K for more details about the ARC-AGI benchmark. An extended survey of other related work is also included in Appendix H.

As of March 2025, the ARC Prize foundation has launched a new dataset and competition, ARC-AGI-2, which is extremely similar in format to ARC-AGI-1. Since the research in this paper predates the launch, this paper focuses solely on ARC-AGI-1, which in this paper we generally refer to as ARC-AGI.

## 3    Method

We propose that MDL can serve as an effective framework for solving ARC-AGI puzzles. In MDL, a more efficient (i.e., lower-bit) compression of a puzzle correlates with a more accurate solution. To solve ARC-AGI puzzles, we design a system that transforms an incomplete puzzle into a completed one—filling in the answers—by finding a compact representation (i.e., short program,) that when run, reproduces the puzzle with any solution. The challenge is to algorithmically obtain this compact program representation, given the puzzle.

Our key innovation is to notice that we can compile a sampling procedure from any continuous random process into a short program, whose program length is very close to the KL divergence of this process relative to some fixed reference process. This particular kind of compilation is made possible by Relative Entropy Coding (REC) [13]. This fact means we can include randomized tensors in a description, and count up their total description lengths as KL divergences which mirror the program length of the compiled sampling procedures. We can even train the tensors with gradient descent to minimize their description lengths as measured by KL terms. Gradient descent then can serve as a description length minimizer in a space of deep learning based programs. Finally, as long as we know that the description length is being minimized and we are able to extract the solution guess, there is no actual need to run REC or compile any sampling procedures in practice.

In standard machine learning lingo, the operations CompressARC actually needs to perform are: (with some simplifications, also see Figure 1)

1. We start at inference time, and we are given an ARC-AGI puzzle to solve. (e.g., puzzle in the diagram below.)

2. We construct a neural network $f$ (see Appendix C) designed for the puzzle's specifics (e.g., number of examples, observed colors). The network takes random normal input $z \sim N(\mu, \Sigma)$, and outputs per-pixel color logit predictions across all the grids, including an answer grid (3 input-output examples, for a total of 6 grids). Importantly, $f_\theta$ is equivariant to common augmentations—such as reordering input-output pairs (including the answer's pair), color permutations, and spatial rotations/reflections.

3. We initialize the network weights $\theta$ and set the parameters $\mu$ and $\Sigma$ for the $z$ distribution.

4. We jointly optimize $\theta$, $\mu$, $\Sigma$ to minimize the sum of cross-entropies over the known grids (5 of them,) ignoring the answer grid. A KL divergence penalty keeps $N(\mu, \Sigma)$ close to $N(0, 1)$, as in a VAE.

5. Since the generated answer grid is stochastic due to the randomness in $z$, we save the answer grids throughout training and choose the most frequently occuring one as our final prediction.

The short program that we would compile the weight $\theta$ and input $z$ distributions into, in trying to minimize the program code length, looks like the following:

```
z = sample_normal(N(0,I), <seed_z>)
weights = <insert weights here>
puzzle_and_solution_logits = neural_net(z, weights)
puzzle_and_solution = sample_categorical(puzzle_and_solution_logits, <seed_error>)
```

where `<seed_z>` and `<seed_error>` are randomization seeds picked by REC to force $z \sim N(\mu, \Sigma)$ and correct final puzzle sampling, with the seeds being approximately $KL(N(\mu, \Sigma)||N(0, I))$ and `CrossEntropyLoss(puzzle_and_solution_logits, true_puzzle, reduction='sum')` bits long, respectively. Our inference-time training setup and chosen loss function serves entirely to shorten the seeds needed by this compiled program, in order to optimize it for Solomoff induction.

Appendix A contains a more elaborate explanation of why we picked this particular program as our candidate shortest program.

# 4  Architecture

We designed our own neural network architecture for decoding the latents $z$ into ARC-AGI puzzles, illustrated in Figure 2. The most important feature of our architecture is it's equivariances, which are symmetry rules dictating that whenever the input $z$ undergoes a transformation, the output ARC-AGI puzzle must also transform the same way. Some example transformations include reordering of input/output pairs, shuffling colors, flips, rotations, and reflections of grids.

The data format of $z$ is what we call a "multitensor", which is a bucket of tensors that each may or may not have certain dimensions such as example, color, height, width dimensions, which transformations can be applied to. All the equivariances can be described in terms of how they change a multitensor. More details on multitensors are in Appendix B

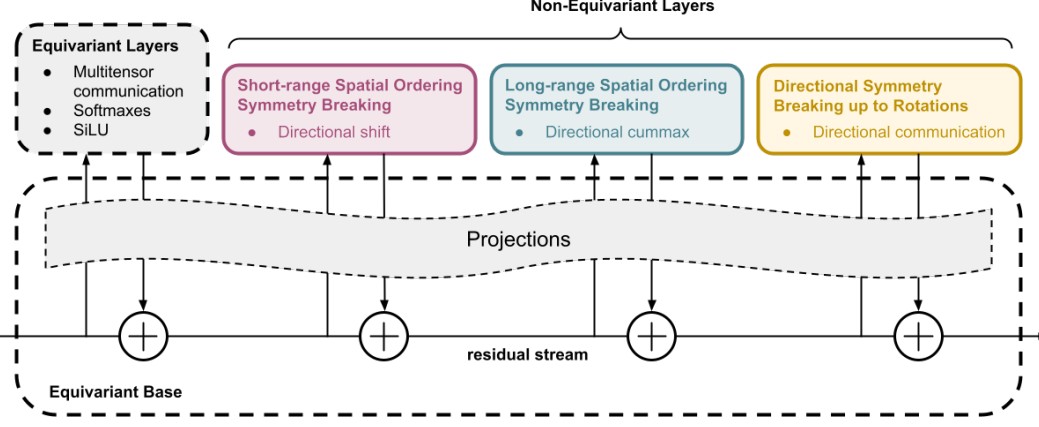

Note: Decoding of z, linear heads, and layernorms are not shown. They are included in the equivariant base.

Figure 2: Overall structure of CompressARC's equivariant neural network. There were too many equivariances for us to consider at once, so we decided to make a **base architecture that's fully symmetric**, and break unwanted symmetries one by one by **adding asymmetric layers** to give it specific non-equivariant abilities (listed later in Appendix G).

The architecture is complicated and has many types of layers that we designed to have inductive biases that are useful for solving the given training puzzles. The training puzzles play no role in our work other than in this way and in our evaluations. The full architecture consists of the following layers, which are each described in the Appendix:

- Begin with parameters of the $z$ distribution
- Decoding Layer, Appendix C.1
- Repeat 4 times:
  - Multitensor Communication Layer (Upwards), Appendix C.2
  - Softmax Layer, Appendix C.3
  - Directional Cummax Layer, Appendix C.4
  - Directional Shift Layer, Appendix C.4
  - Directional Communication Layer, Appendix C.5
  - Nonlinear Layer, Appendix C.6
  - Multitensor Communication Layer (Downwards), Appendix C.2
  - Normalization Layer, Appendix C.7
- Linear Heads, Appendix C.8

# 5   Results

CompressARC solves 20% of evaluation set puzzles and 34.75% of training set puzzles if given 2000 steps per puzzle, as shown in Tables 1 and 2, and Figure 3.

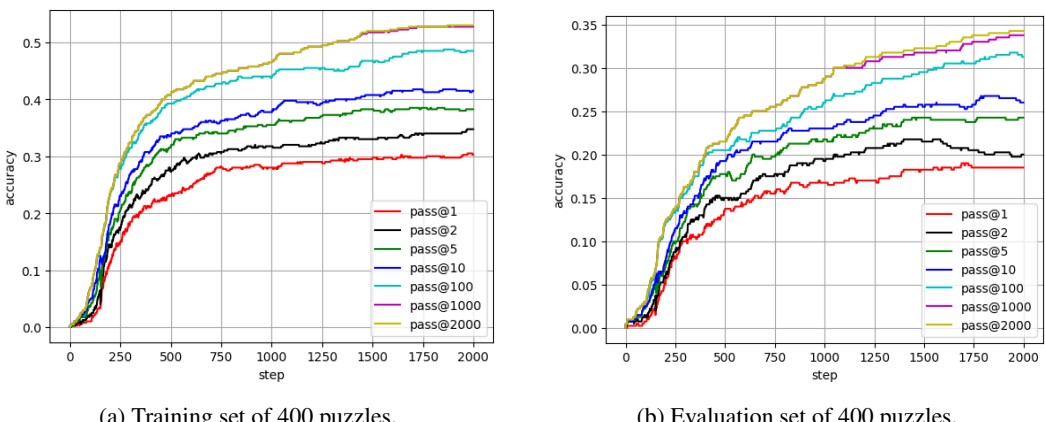

(a) Training set of 400 puzzles.                    (b) Evaluation set of 400 puzzles.

Figure 3: CompressARC's puzzle solve accuracy as a function of the number of steps of inference time learning it is given, for various numbers of allowed attempts (pass@n). The official benchmark is reported with 2 allowed attempts, which is why we report 20% on the evaluation set.

Table 1: CompressARC's puzzle solve accuracy on the training set as a function of the number of steps of inference time learning it is given, for various numbers of allowed attempts (pass@n). The official benchmark is reported with 2 allowed attempts, which is why we report 20% on the evaluation set. Timing is reported for an NVIDIA RTX 4070 GPU.

| Training Iteration | Time | Pass@1 | Pass@2 | Pass@5 | Pass@10 | Pass@100 | Pass@1000 |
|---|---|---|---|---|---|---|---|
| 100 | 6 h | 1.00% | 2.25% | 3.50% | 4.75% | 6.75% | 6.75% |
| 200 | 13 h | 11.50% | 14.25% | 16.50% | 18.25% | 23.25% | 23.50% |
| 300 | 19 h | 18.50% | 21.25% | 23.50% | 26.75% | 31.50% | 32.50% |
| 400 | 26 h | 21.00% | 25.00% | 28.75% | 31.00% | 36.00% | 37.50% |
| 500 | 32 h | 23.00% | 27.50% | 31.50% | 33.50% | 39.25% | 40.75% |
| 750 | 49 h | 28.00% | 30.50% | 34.00% | 36.25% | 42.75% | 44.50% |
| 1000 | 65 h | 28.00% | 31.75% | 35.50% | 37.75% | 43.75% | 46.50% |
| 1250 | 81 h | 29.00% | 32.25% | 37.00% | 39.25% | 45.50% | 49.25% |
| 1500 | 97 h | 29.50% | 33.00% | 38.25% | 40.75% | 46.75% | 51.75% |
| 2000 | 130 h | 30.25% | 34.75% | 38.25% | 41.50% | 48.50% | 52.75% |

Table 2: CompressARC's puzzle solve accuracy on the evaluation set, reported the same way as in Table 1.

| Training Iteration | Time | Pass@1 | Pass@2 | Pass@5 | Pass@10 | Pass@100 | Pass@1000 |
|---|---|---|---|---|---|---|---|
| 100 | 7 h | 0.75% | 1.25% | 2.25% | 2.50% | 3.00% | 3.00% |
| 200 | 14 h | 5.00% | 6.00% | 7.00% | 7.75% | 12.00% | 12.25% |
| 300 | 21 h | 10.00% | 10.75% | 12.25% | 13.25% | 15.50% | 16.25% |
| 400 | 28 h | 11.75% | 13.75% | 16.00% | 17.00% | 19.75% | 20.00% |
| 500 | 34 h | 13.50% | 15.00% | 17.75% | 19.25% | 20.50% | 21.50% |
| 750 | 52 h | 15.50% | 17.75% | 19.75% | 21.50% | 22.75% | 25.50% |
| 1000 | 69 h | 16.75% | 19.25% | 21.75% | 23.00% | 26.00% | 28.75% |
| 1250 | 86 h | 17.00% | 20.75% | 23.00% | 24.50% | 28.25% | 30.75% |
| 1500 | 103 h | 18.25% | 21.50% | 24.25% | 25.50% | 29.50% | 31.75% |
| 2000 | 138 h | 18.50% | 20.00% | 24.25% | 26.00% | 31.25% | 33.75% |

## 5.1 What Puzzles Can and Can't We Solve?

**CompressARC tries to use its abilities to figure out as much as it can, until it gets bottlenecked by one of it's inabilities.**

For example, puzzle 28e73c20 in the training set requires extension of a pattern from the edge towards the middle, as shown in Figure 11a in the Appendix. Given the layers in it's network, CompressARC is generally able to extend patterns for short ranges but not long ranges. So, it does the best that it can, and correctly extends the pattern a short distance before guessing at what happens near the center (Figure 11b, Appendix). Appendix G includes a list of which abilities we have empirically seen CompressARC able to and not able to perform.

## 6 Case Study: Color the Boxes

In this puzzle (Puzzle 272f95fa, Figure 4), you must color sections depending on which side of the grid the section is on. We call this puzzle "Color the Boxes".

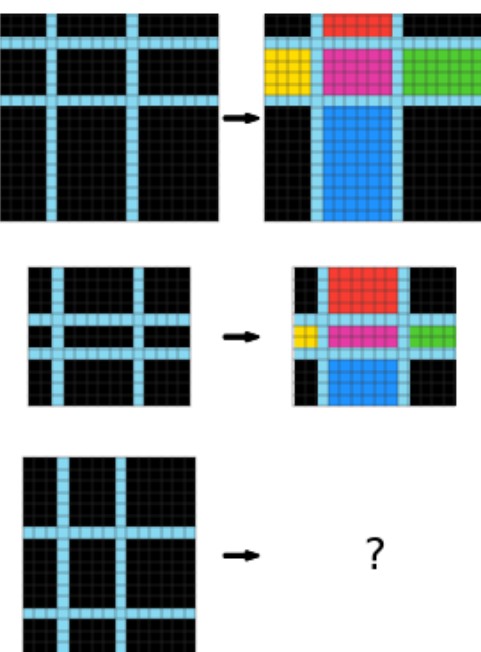

Figure 4: Color the Boxes, problem 272f95fa.

**Human Solution:** We first realize that the input is divided into boxes, and the boxes are still there in the output, but now they're colored. We then try to figure out which colors go in which boxes. First, we notice that the corners are always black. Then, we notice that the middle is always magenta. And after that, we notice that the color of the side boxes depends on which direction they are in: red for up, blue for down, green for right, and yellow for left. At this point, we copy the input over to the answer grid, then we color the middle box magenta, and then color the rest of the boxes according to their direction.

**CompressARC Solution:** Table 3 shows CompressARC's learning behavior over time. After CompressARC is done learning, we can deconstruct it's learned z distribution to find that it codes for a color-direction correspondence table and row/column divider positions (Figure 6).

During training, the reconstruction error fell extremely quickly. It remained low on average, but would spike up every once in a while, causing the KL from $z$ to bump upwards at these moments, as shown in Figure 5a.

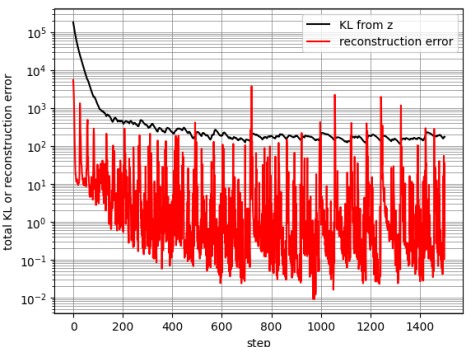

(a) Relative proportion of the KL and reconstruction terms to the loss during training, before taking the weighted sum. The KL dominates the loss and reconstruction is most often nearly perfect.

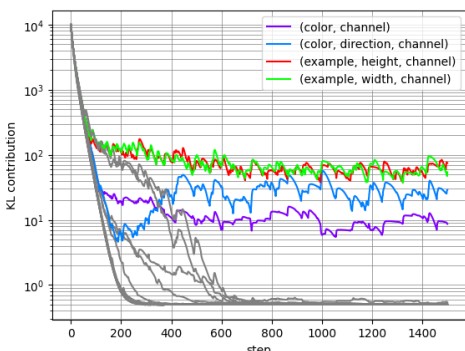

(b) Breaking down the KL loss during training into contributions from each individual shaped tensor in the multitensor $z$. Four tensors dominate, indicating they contain information, and the other 14 fall to zero, indicating their lack of information content.

Figure 5: Breaking down the loss components during training tells us where and how CompressARC prefers to store information relevant to solving a puzzle.

## 6.1 Solution Analysis

So how does CompressARC learn to solve Color the Boxes? We can look at the representations stored in $z$ to find out.

Since $z$ is a multitensor, each of the tensors it contains produces an additive contribution to the total KL for $z$. By looking at the per-tensor contributions (see Figure 5b), we can determine which tensors in $z$ code for information that is used to represent the puzzle.

All the tensors fall to zero information content during training, except for four tensors. In some replications of this experiment, we saw one of these four necessary tensors fall to zero information content, and CompressARC typically does not recover the correct answer after that. Here we are showing a lucky run where the [color, direction, channel] tensor almost falls but gets picked up 200 steps in, which is right around when the samples from the model begin to show the correct colors in the correct boxes.

We can look at the average output of the decoding layer (explained in Appendix C.1) corresponding to individual tensors of $z$, to see what information is stored there (see Figure 6). Each tensor contains a vector of dimension n_channels for various indices of the tensor. Taking the PCA of these vectors reveals some number of activated components, telling us how many pieces of information are coded by the tensor.

Table 3: CompressARC learning the solution for Color the Boxes, over time.

| Learning steps | What is CompressARC doing? | Sampled solution guess |
|---|---|---|
| 50 | CompressARC's network outputs an answer grid (sample) with light blue rows/columns wherever the input has the same. It has noticed that all the other input-output pairs in the puzzle exhibit this correspondence. It doesn't know how the other output pixels are assigned colors; an exponential moving average of the network output (sample average) shows the network assigning mostly the same average color to non-light-blue pixels. | sample    sample average |
| 150 | The network outputs a grid where nearby pixels have similar colors. It has likely noticed that this is common among all the outputs, and is guessing that it applies to the answer too. | sample    sample average |
| 200 | The network output now shows larger blobs of colors that are cut off by the light blue borders. It has noticed the common usage of borders to demarcate blobs of colors in other outputs, and applies the same idea here. It has also noticed black corner blobs in other given outputs, which the network imitates. | sample    sample average |
| 350 | The network output now shows the correct colors assigned to boxes of the correct direction from the center. It has realized that a single color-to-direction mapping is used to pick the blob colors in the other given outputs, so it imitates this mapping. It is still not the best at coloring within the lines, and it's also confused about the center blob, probably because the middle does not correspond to a direction. Nevertheless, the average network output does show a tinge of the correct magenta color in the middle, meaning the network is catching on. | sample    sample average |
| 1500 | The network is as refined as it will ever be. Sometimes it will still make a mistake in the sample it outputs, but this uncommon and filtered out. | sample    sample average |

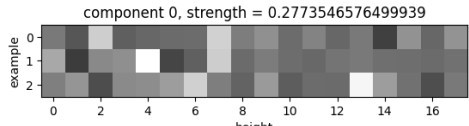

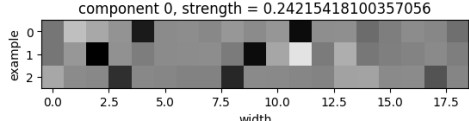

(a) (**example**, **height**, **channel**) **tensor.** For every example and row, there is a vector of dimension n_channels. Taking the PCA of this set of vectors, the top principal component (1485 times stronger than the other components) visualized as the (example, height) matrix shown above tells us which examples/row combinations are uniquely identified by the stored information. **For every example, the two brightest pixels give the rows where the light blue rows in the grids are.**

(b) (**example**, **width**, **channel**) **tensor.** A similar story here to 6a: in the top principal component of this tensor, **the two darkest pixels for every example give the columns where the light blue columns in the grids are.** The top principal component is 1253 times stronger than the next principal component.

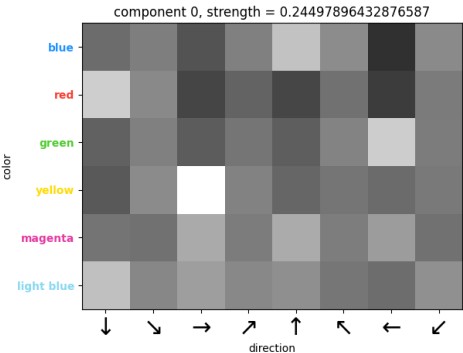

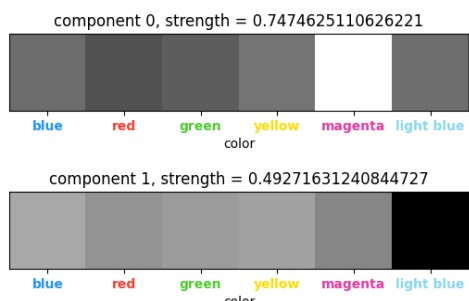

(c) (**direction**, **color**, **channel**) **tensor.** The four brightest pixels identify blue with up, green with left, red with down, and yellow with right. **This tensor tells each direction which color to use for the opposite edge's box.** The top principal component is 829 times stronger than the next principal component.

(d) (**color**, **channel**) **tensor.** Here, we look at the top three principal components, since the first and second principal components are 134 and 87 times stronger than the third component, indicating that they play a role while the third component does not. The **magenta and light blue colors** are uniquely identified, indicating their special usage amongst the rest of the colors as **the center color and the color of the row/column divisions**, respectively.

Figure 6: Breaking down the loss components during training tells us where and how CompressARC prefers to store information relevant to solving a puzzle.

# 7 Discussion

The prevailing reliance of modern deep learning on high-quality data has put the field in a chokehold when applied to problems requiring intelligent behavior that have less data available. This is especially true for the data-limited ARC-AGI benchmark, where LLMs trained on specially augmented, extended, and curated datasets dominate. In the midst of this circumstance, we built CompressARC, which not only uses no training data at all, but forgoes the entire process of pretraining altogether. One should intuitively expect this to fail and solve no puzzles at all, but by applying MDL to the target puzzle during inference time, CompressARC solves a surprisingly large portion of ARC-AGI-1.

CompressARC's theoretical underpinnings come from minimizing the description length of the target puzzle. While other MDL search strategies have been scarce due to the intractablly large search space of possible programs, CompressARC explores a simplified, neural network-based search space through gradient descent. Though CompressARC's architecture is heavily engineered, it's incredible ability to generalize from as low as two demonstration input/output pairs puts it in an entirely new regime of generalization for ARC-AGI.

We challenge the assumption that intelligence must arise from massive pretraining and data, showing instead that clever use of MDL and compression principles can lead to surprising capabilities. We use CompressARC a proof of concept to demonstrate that modern deep learning frameworks can be melded with MDL to create a possible alternative, complimentary route to AGI.

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

# A Optimality of Our Candidate Shortest Program

It isn't obvious how we get from trying to minimize the description length to the method we ended up using. The derivation of our algorithm takes us on a detour through information theory [14], algorithmic information theory [15], and coding theory [16], with machine learning only making an appearance near the end.

## A.1 A Primer on Lossless Information Compression

In information theory, lossless information compression is about trying to represent some information in as few bits as possible, while still being able to reconstruct that information from the bit representation. [17] This type of problem is abstracted as follows:

- A source produces some symbol $x$ from some process that generates symbols from a probability distribution $p(x)$.
- A compressor/encoder $E$ must map the symbol $x$ to a string of bits $s$.
- A decompressor/decoder $D$ must exactly map $s$ back to the original symbol $x$.

The goal in lossless information compression is to use $p$ to construct functions $(E, D)$ which are bit-efficient, (i.e., that minimize the expected length of $s$,) without getting any symbols wrong. The optimal decompressor $D^*$ also plays a role in a program that is the shortest possible (up to additive constants in program length) that computes $x$, in expectation over $x$ drawn from $p$:

```
s = <string of bits>
x = D*(s)
```

This reduces MDL to the problem of lossless information compression. In our case, the symbol $x$ is the ARC-AGI dataset (many puzzle + answer pairs), and we may want to figure out what D* is using knowledge of $p$, and what s is when given $x$. Except, we won't have the answers (only the puzzles) in $x$, and we don't actually know $p$, since it's hard to model the intelligent process of puzzle ideation in humans.

## A.2 One-Size-Fits-All Compression

To build an efficient lossless compression scheme, you might think we need to know what $p$ is, but we argue that it doesn't really matter since we can make a one-size-fits-all compressor. It all hinges on the following assumption:

**There exists some practically implementable, bit efficient compression system $(E, D)$ for ARC-AGI datasets $x$ sampled from $p$.**

If this were false, our whole idea of solving ARC-AGI with compression will be doomed even if we knew $p$ anyways, so we might as well make this assumption.

Our one-size-fits-all compressor $(E', D')$ is built without knowing $p$, and it is almost just as bit-efficient as the original $(E, D)$:

- $E'$ observes symbol $x$, picks a program $f$ and input $s$ to minimize $\text{len}(f) + \text{len}(s)$ under the constraint that running the program makes $f(s) = x$, and then sends the pair $(f, s)$.
- $D'$ is just a program executor that executes $f$ on $s$, correctly producing $x$.

It is possible to prove with algorithmic information theory that $(E', D')$ achieves a bit efficiency at most $\text{len}(f)$ bits worse than the bit efficiency of $(E, D)$, where $f$ is the code for implementing $D$. [15] But since compression is practically implementable, the code for $D$ should be simple enough for a human engineer to write, so $\text{len}(f)$ must be short, meaning our one-size-fits-all compressor will be close to the best possible bit efficiency.

Ironically, the only problem with using this to solve ARC-AGI is that implementing $E'$ is not practical, since $E'$ needs to minimize the length of a program-input pair $(f, s)$ under partial fixed output constraint $f(s)_{\text{puzzle}} = x_{\text{puzzle}}$.

### A.3 Neural Networks to the Rescue

To avoid searching through program space, we just pick a program $f$ for a small sacrifice in bit efficiency. We hope the diversity of program space can be delegated to diversity in input $s$ space instead. Specifically, we write a program $f$ that runs the forward pass of a neural network, where $s = (\theta, z, \epsilon)$ are the weights, inputs, and corrections to the outputs of the neural network. Then, we can use gradient descent to "search" over $s$.

This restricted compression scheme uses Relative Entropy Coding (REC) [13][1] to encode noisy weights $\theta$ and neural network inputs $z$ into bits $s_\theta$ and $s_z$, and arithmetic coding [18] to encode output error corrections $\epsilon$ into bits $s_\epsilon$, to make a bit string $s$ consisting of three blocks $(s_\theta, s_z, s_\epsilon)$. The compression scheme runs as follows:

- The decoder runs $\theta = \text{REC-decode}(s_\theta)$, $z = \text{REC-decode}(s_z)$, logits = Neural-Net$(\theta, z)$, and $x = \text{Arithmetic-decode}(s_\epsilon, \text{logits})$.

- The encoder trains $\theta$ and $z$ to minimize the total code length $\mathbb{E}[\text{len}(s)]$. $s_\epsilon$ is fixed by arithmetic coding to guarantee correct decoding. To calculate the three components of the loss $\mathbb{E}[\text{len}(s)]$ in a differentiable way, we refer to the properties of REC and arithmetic coding:

  - It turns out that the $\epsilon$ code length $\mathbb{E}[\text{len}(s_\epsilon)]$ is equal to the total crossentropy error on all the given grids in the puzzle.
  - REC requires us to fix some reference distribution $q_\theta$, and also add noise to $\theta$, turning it into a distribution $p_\theta$. Then, REC allows you to store noisy $\theta$ using a code length of $\mathbb{E}[\text{len}(s_\theta)] = \text{KL}(p_\theta||q_\theta) = \mathbb{E}_{\theta \sim p_\theta}[\log(p_\theta(\theta)/q_\theta(\theta))]$ bits. We will choose to fix $q_\theta = N(0, I/2\lambda)$ for large $\lambda$, such that the loss component $\mathbb{E}[\text{len}(s_\theta)] \approx \lambda|\theta|^2 + \text{const}$ is equivalent to regularizing the decoder.
  - We must also do for $z$ what we do for $\theta$, since it's also represented using REC. We will choose to fix $q_z = N(0, I)$, so the code length of $z$ is $\mathbb{E}[\text{len}(s_z)] = \text{KL}(p_z||q_z) = \mathbb{E}_{z \sim p_z}[\log(p_z(z)/q_z(z))]$.

  We can compute gradients of these code lengths via the reparameterization trick. [19]

At this point, we observe that the total code length for $s$ that we described is actually the VAE loss with decoder regularization (= KL for $z$ + reconstruction error + regularization).[2] Likewise, if we port the rest of what we described above (plus modifications regarding equivariances and inter-puzzle independence, and ignoring regularization) into typical machine learning lingo, we get the previous description of CompressARC from Section 3.

## B  Multitensors

The actual data ($z$, hidden activations, and puzzles) passing through our layers comes in a format that we call a "**multitensor**", which is just a bucket of tensors of various shapes, as shown in Figure 7. All the equivariances we use can be described in terms of how they change a multitensor.

Most common classes of machine learning architectures operate on a single type of tensor with constant rank. LLMs operate on rank-3 tensors of shape $[\text{n\_batch}, \text{n\_tokens}, \text{n\_channels}]$, and Convolutional Neural Networks (CNNs) operate on rank-4 tensors of shape

---

[1]A lot of caveats/issues are introduced by using REC. The code length when using REC only behaves in some limits and expectations, there may be a small added constant to the code length, the decoding may be approximate, etc. We're not up to date with the current literature, and we're ignoring all the sticky problems that may arise and presuming that they are all solved. We will never end up running Relative Entropy Coding anyways, so it doesn't matter that it takes runtime exponential in the code length. We only need to make use of the the fact that such algorithms exist, not that they run fast, nor that we can implement them, in order to derive our method.

[2]We penalize the reconstruction error by 10x the KL for $z$, in the total KL loss. This isn't detrimental to the measurement of the total KL because the KL term for $z$ can absorb all of the coded information from the reconstruction term, which can then go to zero. Since the term for $z$ is not penalized by any extra factor, the total KL we end up with is then unaffected. We believe this empirically helps because the Gaussians we use for $z$ are not as efficient for storing bits that can be recovered, as the categorical distributions that define the log likelihood in the reconstruction error. Forcing all the coded bits into one storage mode removes pathologies introduced by multiple storage modes.

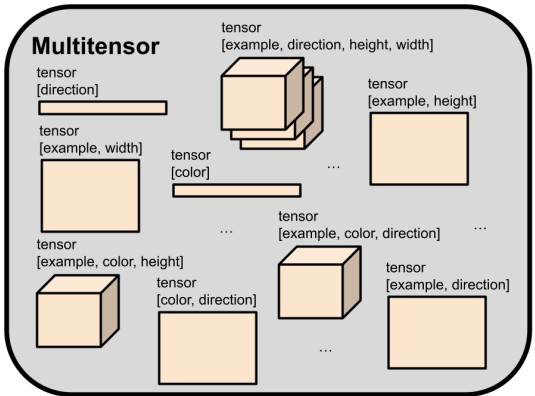

Note: channel dimension not shown.

Figure 7: Our neural network's internal representations come in the form of a "multitensor", a bucket of tensors of different shapes. One of the tensors is shaped like $[example, color, height, width, channel]$, an adequate shape for storing a whole ARC-AGI puzzle.

$[n\_batch, n\_channels, height, width]$. Our multitensors are a set of varying-rank tensors of unique type, whose dimensions are a subset of a rank-6 tensor of shape $[n\_examples, n\_colors, n\_directions, height, width, n\_channels]$, as illustrated in Figure 7. We always keep the channel dimension, so there are at most 32 tensors in each multitensor. We also maintain several rules (see Appendix D.1) that determine whether a tensor shape is "legal" or not, which reduces the number of tensors in a multitensor to 18.

| Dimension | Description |
|---|---|
| Example | Number of examples in the ARC-AGI puzzle, including the one with held-out answer |
| Color | Number of unique colors in the ARC-AGI puzzle, not including black, see Appendix E.2 |
| Direction | 8 |
| Height | Determined when preprocessing the puzzle, see Appendix E.1 |
| Width | Determined when preprocessing the puzzle, see Appendix E.1 |
| Channel | In the residual connections, the size is 8 if the direction dimension is included, else 16. Within layers it is layer-dependent. |

Table 4: Size conventions for multitensor dimensions.

To give an idea of how a multitensor stores data, an ARC-AGI puzzle can be represented by using the $[example, color, height, width, channel]$ tensor, by using the channel dimension to select either the input or output grid, and the height/width dimensions for pixel location, a one hot vector in the color dimension, specifying what color that pixel is. The $[example, height, channel]$ and $[example, width, channel]$ tensors can similarly be used to store masks representing grid shapes for every example for every input/output grid. All those tensors are included in a single multitensor that is computed by the network just before the final linear head (described in Appendix C.8).

When we apply an operation on a multitensor, we by default assume that all non-channel dimensions are treated identically as batch dimensions by default. The operation is copied across the indices of dimensions unless specified. This ensures that we keep all our symmetries intact until we use a specific layer meant to break a specific symmetry.

A final note on the channel dimension: usually when talking about a tensor's shape, we will not even mention the channel dimension as it is included by default.

# C  Layers in the Architecture

## C.1  Decoding Layer

This layer's job is to sample a multitensor $z$ and bound its information content, before it is passed to the next layer. This layer and outputs the KL divergence between the learned $z$ distribution and $N(0, I)$. Penalizing the KL prevents CompressARC from learning a distribution for $z$ that memorizes the ARC-AGI puzzle in an uncompressed fashion, and forces CompressARC to represent the puzzle more succinctly. Specifically, it forces the network to spend more bits on the KL whenever it uses $z$ to break a symmetry, and the larger the symmetry group broken, the more bits it spends.

This layer takes as input:

- A learned target multiscalar, called the "target capacity".[3] The decoding layer will output $z$ whose information content per tensor is close to the target capacity,[4]
- learned per-element means for $z$,[5]
- learned per-element capacity adjustments for $z$.

We begin by normalizing the learned per-element means for $z$.[6] Then, we figure out how much Gaussian noise we must add into every tensor to make the AWGN channel capacity [17] equal to the target capacity for every tensor (including per-element capacity adjustments). We apply the noise to sample $z$, keeping unit variance of $z$ by rescaling.[7]

We compute the information content of $z$ as the KL divergence between the distribution of this sample and $N(0, 1)$.

Finally, we postprocess the noisy $z$ by scaling it by the sigmoid of the signal-to-noise ratio.[8] This ensures that $z$ is kept as-is when its variance consists mostly of useful information and it is nearly zero when its variance consists mostly of noise. All this is done 4 times to make a channel dimension of 4. Then we apply a projection (with different weights per tensor in the multitensor, i.e., per-tensor projections) mapping the channel dimension up to the dimension of the residual stream.

## C.2  Multitensor Communication Layer

This layer allows different tensors in a multitensor to interact with each other.

First, the input from the residual stream passes through per-tensor projections to a fixed size (8 for downwards communication and 16 for upwards communication). Then a message is sent to every other tensor that has at least the same dimensions for upwards communication, or at most the same dimensions for downwards communication. This message is created by either taking means along dimensions to remove them, or unsqueezing+broadcasting dimensions to add them, as in Figure 8. All the messages received by every tensor are summed together and normalization is applied. This result gets up-projected back and then added to the residual stream.

## C.3  Softmax Layer

This layer allows the network to work with internal one-hot representations, by giving it the tools to denoise and sharpen noisy one-hot vectors. For every tensor in the input multitensor, this layer lists out all the possible subsets of dimensions of the tensor to take a softmax over,[9] takes the softmax

---

[3]Target capacities are exponentially parameterized and rescaled by 10x to increase sensitivity to learning, initialized at a constant $10^4$ nats per tensor, and forced to be above a minimum value of half a nat.

[4]The actual information content, which the layer computes later on, will be slightly different because of the per-element capacity adjustments.

[5]Means are initialized using normal distribution of variance $10^{-4}$.

[6]Means and variances for normalization are computed along all non-channel dimensions.

[7]There are many caveats with the way this is implemented and how it works; please refer to the code (see Appendix N) for more details.

[8]We are careful not to let the postprocessing operation, which contains unbounded amounts of information via the signal-to-noise ratios, to leak lots of information across the layer. We only let a bit of it leak by averaging the signal-to-noise ratios across individual tensors in the multitensor.

[9]One exception: we always include the example dimension in the subset of dimensions.

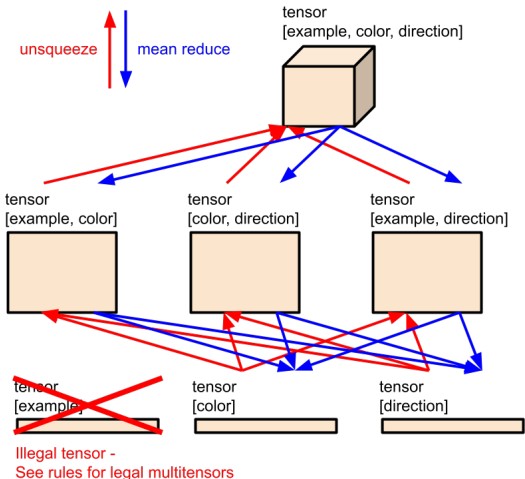

Figure 8: Multitensor communication layer. Higher rank tensors shown at the top, lower rank at the bottom. Tensors transform between ranks by mean reduction and unsqueezing dimensions.

over these subsets of dimensions, and concatenates all the softmaxxed results together in the channel dimension. The output dimension varies across different tensors in the multitensor, depending on their tensor rank. A pre-norm is applied, and per-tensor projections map to and from the residual stream. The layer has input channel dimension of 2.

## C.4 Directional Cummax/Shift Layer

The directional cummax and shift layers allow the network to perform the non-equivariant cummax and shift operations in an equivariant way, namely by applying the operations once per direction, and only letting the output be influenced by the results once the directions are aggregated back together (by the multitensor communication layer). These layers are the sole reason we included the direction dimension when defining a multitensor: to store the results of directional layers and operate on each individually. Of course, this means when we apply a spatial equivariance transformation, we must also permute the indices of the direction dimension accordingly, which can get complicated sometimes.

The directional cummax layer takes the eight indices of the direction dimension, treats each slice as corresponding to one direction (4 cardinal, 4 diagonal), performs a cumulative max in the respective direction for each slice, does it in the opposite direction for half the channels, and stacks the slices back together in the direction dimension. An illustration is in Figure 9. The slices are rescaled to have min $-1$ and max 1 before applying the cumulative max.

The directional shift layer does the same thing, but for shifting the grid by one pixel instead of applying the cumulative max, and without the rescaling.

Some details:

• Per-tensor projections map to and from the residual stream, with pre-norm.

• Input channel dimension is 4.

• These layers are only applied to the $[example, color, direction, height, width, channel]$ and $[example, direction, height, width, channel]$ tensors in the input multitensor.

## C.5 Directional Communication Layer

By default, the network is equivariant to permutations of the eight directions, but we only want symmetry up to rotations and flips. So, this layer provides a way to send information between two slices in the direction dimension, depending on the angular difference in the two directions. This layer defines a separate linear map to be used for each of the 64 possible combinations of angles, but the weights of the linear maps are minimally tied such that the directional communication layer

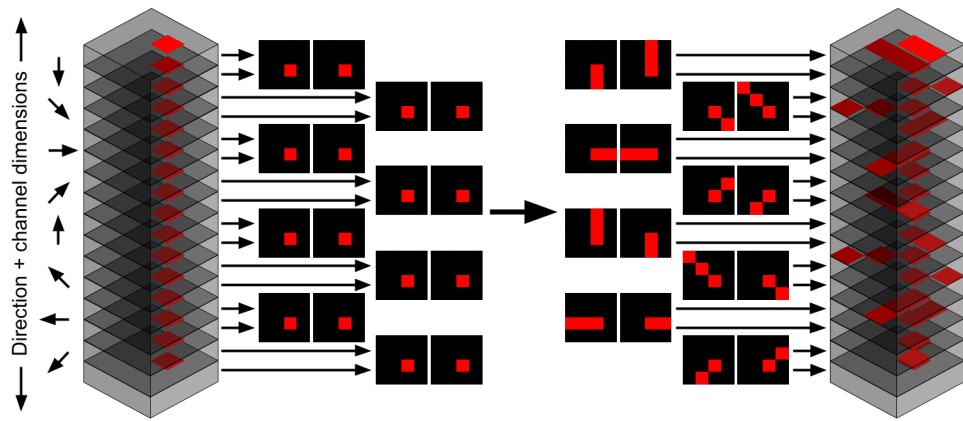

Figure 9: The directional cummax layer takes a directional tensor, splits it along the direction axis, and applies a cumulative max in a different direction for each direction slice. This operation helps CompressARC transport information across long distances in the puzzle grid.

is equivariant to reflections and rotations. This gets complicated really fast, since the direction dimension's indices also permute when equivariance transformations are applied. Every direction slice in a tensor accumulates it's 8 messages, and adds the results together.[10]

For this layer, there are per-tensor projections to and from the residual stream with pre-norm. The input channel dimension is 2.

### C.6    Nonlinear Layer

We use a SiLU nonlinearity with channel dimension 16, surrounded by per-tensor projections with pre-norm.

### C.7    Normalization Layer

We normalize all the tensors in the multitensor, using means and variances computed across all dimensions except the channel dimension. Normalization as used within other layers also generally operates this way.

### C.8    Linear Heads

We must take the final multitensor, and convert it to the format of an ARC-AGI puzzle. More specifically, we must convert the multitensor into a distribution over ARC-AGI puzzles, so that we can compute the log-likelihood of the observed grids in the puzzle.

The colors of every pixel for every example for both input and output, have logits defined by the [example, color, height, width, channel] tensor, with the channel dimension linearly mapped down to a size of 2, representing the input and output grids.[11] The log-likelihood is given by the crossentropy, with sum reduction across all the grids.

For grids of non-constant shape, the [example, height, channel] and [example, width, channel] tensors are used to create distributions over possible contiguous rectangular slices of each grid of colors, as shown in Figure 10. Again, the channel dimension is mapped down to a size of 2 for input and output grids. For every grid, we have a vector of size [width] and a vector of size [height]. The log

---

[10]We also multiply the results by coefficients depending on the angle: 1 for 0 degrees and 180 degrees, 0.2 for 45 degrees and 135 degrees, and 0.4 for 90 degrees.

[11]The linear map is initialized to be identical for both the input and output grid, but isn't fixed this way during learning. Sometimes this empirically helps with problems of inconsistent input vs output grid shapes. The bias on this linear map is multiplied by 100 before usage, otherwise it doesn't seem to be learned fast enough empirically. This isn't done for the shape tensors described by the following paragraph though.

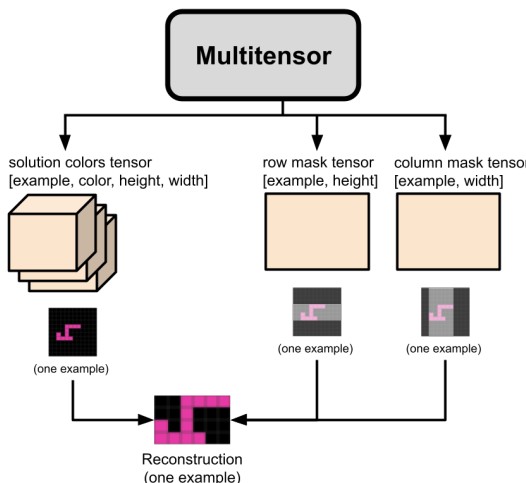

Figure 10: The linear head layer takes the final multitensor of the residual stream and reads a [example, color, height, width, channel] tensor to be interpreted as color logits, and a [example, height, channel] tensor and a [example, width, channel] tensor to serve as shape masks.

likelihood of every slice of the vector is taken to be the sum of the values within the slice, minus the values outside the slice. The log likelihoods for all the possible slices are then normalized to have total probability one, and the colors for every slice are given by the color logits defined in the previous paragraph.

With the puzzle distribution now defined, we can now evaluate the log-likelihood of the observed target puzzle, to use as the reconstruction error.[12]

# D    Other Architectural Details

## D.1    Rules for legal multitensors

1. At least one non-example dimension must be included. Examples are not special for any reason not having to do with colors, directions, rows, and columns.

2. If the width or height dimension is included, the example dimension should also be included. Positions are intrinsic to grids, which are indexed by the example dimension. Without a grid it doesn't make as much sense to talk about positions.

## D.2    Weight Tying for Reflection/Rotation Symmetry

When applying a different linear layer to every tensor in a multitensor, we have a linear layer for tensors having a width but not height dimension, and another linear layer for tensors having a height but not width dimension. Whenever this is the case, we tie the weights together in order to preserve the whole network's equivariance to diagonal reflections and 90 degree rotations, which swap the width and height dimensions.

The softmax layer is not completely symmetrized because different indices of the output correspond to different combinations of dimension to softmax over. Tying the weights properly would be a bit complicated and time consuming for the performance improvement we expect, so we did not do this.

---

[12]There are multiple slices of the same shape that result in the correct puzzle to be decoded. We sum together the probabilities of getting any of the slices by applying a logsumexp to the log probabilities. But, we found empirically that training prematurely collapses onto one particular slice. So, we pre-multiply and post-divide the log probabilities by a coefficient when applying the logsumexp. The coefficient starts at 0.1 and increases exponentially to 1 over the first 100 iterations of training. We also pre-multiply the masks by the square of this coefficient as well, to ensure they are not able to strongly concentrate on one slice too early in training.

 **D.3  Training**

 We train for 2000 iterations using Adam, with learning rate 0.01, $\beta_1$ of 0.5, and $\beta_2$ of 0.9.

# E   Preprocessing

## E.1   Output Shape Determination

The raw data consists of grids of various shapes, while the neural network operates on grids of constant shape. Most of the preprocessing that we do is aimed towards this shape inconsistency problem.

Before doing any training, we determine whether the given ARC-AGI puzzle follows three possible shape consistency rules:

1. The outputs in a given ARC-AGI puzzle are always the same shape as corresponding inputs.

2. All the inputs in the given ARC-AGI puzzle are the same shape.

3. All the outputs in the given ARC-AGI puzzle are the same shape.

Based on rules 1 and 3, we try to predict the shape of held-out outputs, prioritizing rule 1 over rule 3. If either rule holds, we force the postprocessing step to only consider the predicted shape by overwriting the masks produced by the linear head layer. If neither rule holds, we make a temporary prediction of the largest width and height out of the grids in the given ARC-AGI puzzle, and we allow the masks to predict shapes that are smaller than that.

The largest width and height that is given or predicted, are used as the size of the multitensor's width and height dimensions.

The predicted shapes are also used as masks when performing the multitensor communication, directional communication and directional cummax/shift layers. We did not apply masks for the other layers because of time constraints and because we do not believe it will provide for much of a performance improvement.[13]

## E.2   Number of Colors

We notice that in almost all ARC-AGI puzzles, colors that are not present in the puzzle are not present in the true answers. Hence, any colors that do not appear in the puzzle are not given an index in the color dimension of the multitensor.

In addition, black is treated as a special color that is never included in the multitensor, since it normally represents the background in many puzzles. When performing color classification, a tensor of zeros is appended to the color dimension after applying the linear head, to represent logits for the black color.

# F   Postprocessing

Postprocessing primarily deals with denoising the answers sampled from the network. This is complicated by the variable shape grids present in some puzzles.

Generally, when we sample answers from the network by taking the logits of the [example, color, height, width, channel] tensor and argmaxxing over the color dimension, we find that the grids are noisy and will often have the wrong colors for several random pixels. We developed several methods for removing this noise:

1. Find the most commonly sampled answer.

2. Construct an exponential moving average of the output color logits before taking the softmax to produce probabilities. Also construct an exponential moving average of the masks.

---

[13]The two masks for the input and output are combined together to make one mask for use in these operations, since the channel dimension in these operations don't necessarily correspond to the input and output grids.

3. Construct an exponential moving average of the output color probabilities after taking the softmax. Also construct an exponential moving average of the masks.

When applying these techniques, we always take the slice of highest probability given the mask, and then we take the colors of highest probability afterwards.

We explored several different rules for when to select which method, and arrived at a combination of 1 and 2 with a few modifications:

- At every iteration, count up the sampled answer, as well as the exponential moving average answer (decay $= 0.97$).
- If before 150 iterations of training, then downweight the answer by a factor of $e^{-10}$. (Effectively, don't count the answer.)
- If the answer is from the exponential moving average as opposed to the sample, then downweight the answer by a factor of $e^{-4}$.
- Downweight the answer by a factor of $e^{-10*\text{uncertainty}}$, where uncertainty is the average (across pixels) negative log probability assigned to the top color of every pixel.

# G   Empirically Observed Abilities and Disabilities of CompressARC

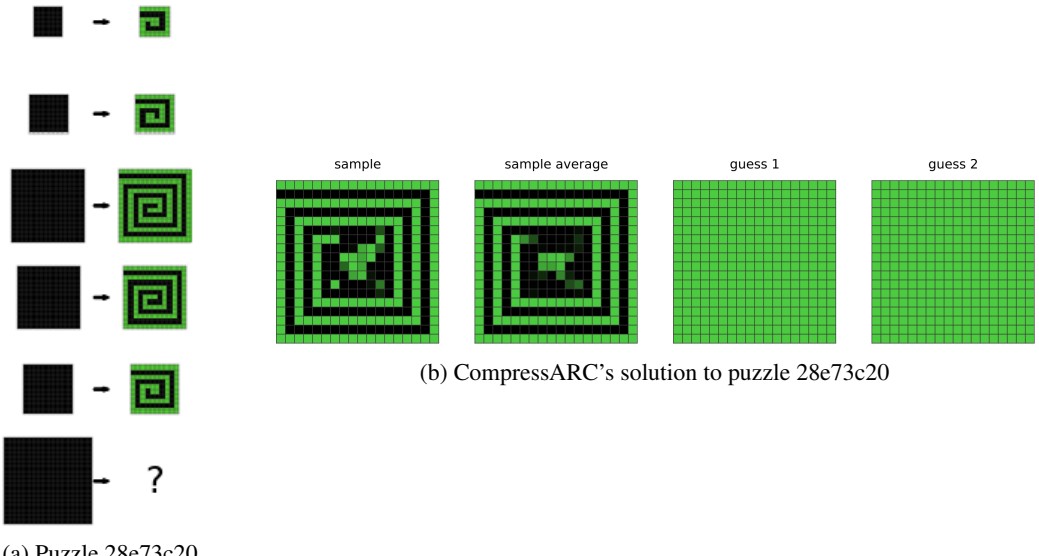

(a) Puzzle 28e73c20

(b) CompressARC's solution to puzzle 28e73c20

Figure 11: Puzzle 28e73c20, and CompressARC's solution to it.

A short list of abilities that **can** be performed by CompressARC includes:

- Assigning individual colors to individual procedures (see puzzle 0ca9ddb6)
- Infilling (see puzzle 0dfd9992)
- Cropping (see puzzle 1c786137)
- Connecting dots with lines, including 45 degree diagonal lines (see puzzle 1f876c06)
- Same color detection (see puzzle 1f876c06)
- Identifying pixel adjacencies (see puzzle 42a50994)
- Assigning individual colors to individual examples (see puzzle 3bd67248)
- Identifying parts of a shape (see puzzle 025d127b)
- Translation by short distances (see puzzle 025d127b)

We believe these abilities to be individually endowed by select layers in the architecture, which we designed specifically for the purpose of conferring those abilities to CompressARC.

A short list of abilities that **cannot** be performed by CompressARC includes:

- Assigning two colors to each other (see puzzle 0d3d703e)
- Repeating an operation in series many times (see puzzle 0a938d79)
- Counting/numbers (see puzzle ce9e57f2)
- Translation, rotation, reflections, rescaling, image duplication (see puzzles 0e206a2e, 5ad4f10b, and 2bcee788)
- Detecting topological properties such as connectivity (see puzzle 7b6016b9)
- Planning, simulating the behavior of an agent (see puzzle 2dd70a9a)
- Long range extensions of patterns (see puzzle 28e73c20 above)

# H  Related Work

## H.1  Equivalence of Compression and Intelligence

The original inspiration of this work came from the Hutter Prize [20], which awards a prize for those who can compress a file of Wikipedia text the most, as a motivation for researchers to build intelligent systems. It is premised upon the idea that the ability to compress information is equivalent to intelligence.

This equivalence between intelligence and compression has a long history. For example, when talking about intelligent solutions to prediction problems, the ideal predictor implements Solomonoff Induction, a theoretically best possible but uncomputable prediction algorithm that works universally for all prediction tasks. [21] This prediction algorithm is then equivalent to a best possible compression algorithm whose compressed code length is the Kolmogorov Complexity of the data. [6] This prediction algorithm can also be used to decode a description of the data of minimal length, linking these formulations of intelligence to MDL. [5] In our work, we try to approximate this best possible compression algorithm with a neural network.

## H.2  Information Theory and Coding Theory

Since we build an information compression system, we make use of many results in information theory and coding theory. The main result required to motivate our model architecture is the existence of Relative Entropy Coding (REC). [13] The fact that REC exists means that as long as a KL divergence can be bounded, the construction of a compression algorithm is always possible and the issue of realizing the algorithm can be abstracted away. Thus, problems about coding theory and translating information from Gaussians into binary and back can be ignored, since we can figure out the binary code length directly from the Gaussians instead. In other words, we only need to do enough information theory using the Gaussians to get the job done, with no coding theory at all. While the existence of arithmetic coding [18] would suffice to abstract the problem away when distributions are discrete, neural networks operate in a continuous space so we need REC instead.

Our architecture sends $z$ information through an additive white Gaussian noise (AWGN) channel, so the AWGN channel capacity formula (Gaussian input Gaussian noise) plays a heavy role in the design of our decoding layer. [17]

## H.3  Variational Autoencoders

The decoder side of the variational autoencoder [19] serves as our decompression algorithm. While we would use something that has more general capabilities like a neural Turing machine [22] instead, neural Turing machines are not very amenable to gradient descent-based optimization so we stuck with the VAE.

VAEs have a long history of developments that are relevant to our work. At one point, we tried using multiple decoding layers to make a hierarchical VAE decoder [23] instead. This does not affect the KL calculation because a channel capacity with feedback is equal to the channel capacity without

feedback. [24] But, we found empirically that the first decoding layer would absorb all of the KL contribution, making the later decoding layers useless. Thus, we only used one decoding layer at the beginning.

The beta-VAE [25] introduces a reweighting of the reconstruction loss to be stronger than the KL loss, and we found that to work well in our case. The NVAE applies a non-constant weighting to loss components. [26] A rudimentary form of scheduled loss recombination is used in CompressARC.

### H.4 ARC-AGI Methods

Aside from LLM-based methods for solving ARC with data augmentation, synthetic datasets, fine-tuning, test-time training, and reasoning, several other classes of solution have been studied:

- An older class of methods consists of hard-coded searches through program spaces in hand-written domain-specific languages designed specifically for ARC. [27, 28]
- [29] introduced a VAE-based method for searching through a latent space of programs. This is the most similar work to ours that we found due to their VAE setup.

### H.5 Deep Learning Architectures

We designed our own neural network architecture from scratch, but not without borrowing crucial design principles from many others.

Our architecture is fundamentally structured like a transformer, consisting of a residual stream where representations are stored and operated upon, followed by a linear head. [30, 31] Pre-and post-norms with linear up- and down-projections allow layers to read and write to the residual stream. [32] The SiLU-based nonlinear layer is especially similar to a transformer's. [33]

Our equivariance structures are inspired by permutation-invariant neural networks, which are a type of equivariant neural network. [34, 35] Equivariance transformations are taken from common augmentations to ARC-AGI puzzles.

# I How to Improve Our Work

At the time of release of CompressARC, there were several ideas which we thought of trying or attempted at some point, but didn't manage to get working for one reason or another. Some ideas we still believe in, but didn't use, are listed below.

### I.1 Joint Compression via Weight Sharing Between Puzzles

CompressARC tries to solve each puzzle serially by compressing each puzzle on its own. We believe that joint compression of all the entire ARC-AGI dataset at once should yield better learned inductive biases per-puzzle, since computations learned for one puzzle can be transferred to other puzzles. We do not account for the complexity of $f$ in our derivation of CompressARC, allowing for $f$ to be used for memorization/overfitting. By jointly compressing the whole dataset, we only need to have one $f$, whereas when compressing each puzzle individually, we need to have an $f$ for every puzzle, allowing for more memorization/overfitting.

To implement this, we would most likely explore strategies like:

- Using the same network weights for all puzzles, and training for puzzles in parallel. Each puzzle gets assigned some perturbation to the weights, that is constrained in some way, e.g., LORA. [36]
- Learning a "puzzle embedding" for every puzzle that is a high dimensional vector (more than 16 dim, less than 256 dim), and learning a linear mapping from puzzle embeddings to weights for our network. This mapping serves as a basic hypernetwork, i.e., a neural network that outputs weights for another neural network. [37]

In a successful case, we might want to also try adding in some form of positional encodings, with the hope that $f$ is now small/simple enough to be incapable of memorization/overfitting using positional encodings.

678 The reason we didn't try this is because it would slow down the research iteration process.

## I.2 Convolution-like Layers for Shape Copying Tasks

680 This improvement is more ARC-AGI-specific and may have less to do with AGI in our view. Many
681 ARC-AGI-1 puzzles can be seen to involve copying shapes from one place to another, and our
682 network has no inductive biases for such an operation. An operation which is capable of copying
683 shapes onto multiple locations is the convolution. With one grid storing the shape and another with
684 pixels activated at locations to copy to, convolving the two grids will produce another grid with the
685 shape copied to the designated locations.

686 There are several issues with introducing a convolutional operation for the network to use. Ideally,
687 we would read two grids via projection from the residual stream, convolve them, and write it back in
688 via another projection, with norms in the right places and such. Ignoring the fact that the grid size
689 changes during convolution (can be solved with two parallel networks using different grid sizes), the
690 bigger problem is that convolutions tend to amplify noise in the grids much more than the sparse
691 signals, so their inductive bias is not good for shape copying. We can try to apply a softmax to one
692 or both of the grids to reduce the noise (and to draw an interesting connection to attention), but we
693 didn't find any success.

694 The last idea that we were tried before discarding the idea was to modify the functional form of the
695 convolution:

$$(f * g)(x) = \sum_y f(x - y)g(y)$$

696 to a tropical convolution [38], which we found to work well on toy puzzles, but not well enough for
697 ARC-AGI-1 training puzzles (which is why we discarded this idea):

$$(f * g)(x) = \max_y f(x - y) + g(y)$$

698 Convolutions, when repeated with some grids flipped by 180 degrees, tend to create high activations
699 at the center pixel, so sometimes it is important to zero out the center pixel to preserve the signal.

## I.3 KL Floor for Posterior Collapse

701 We noticed during testing that crucial posterior tensors whose KL fell to zero during learning would
702 never make a recovery and play their role in the encoding, just as in the phenomenon of mode collapse
703 in variational autoencoders. [39] We believe that the KL divergence may upper bound the information
704 content of the gradient training signal for parts of the network that process the encoded information.
705 Thus, when a tensor falls to zero KL, the network stops learning to use its information, so the KL is
706 no longer given encouragement to recover. If we can hold the KL above zero for a while, the network
707 may then learn to use the information, giving the KL a reason to stay above zero when released again.

708 We implemented a mechanism to keep the KL above a minimum threshold so that the network always
709 learns to use that information, but we do not believe it learns fast enough for this to be useful, as we
710 have never seen a tensor recover before. Therefore, it might be useful to explore different ways to
711 schedule this KL floor to start high and decay to zero, to allow learning when the KL is forced to be
712 high, and to leave the KL unaffected later on in learning. This might cause training results to be more
713 consistent across runs.

## I.4 Regularization

715 CompressARC does not use regularization on the weights. Regularization measures the complexity
716 of $f$ in our problem framing, and is native to our derivation of CompressARC in Appendix A.3. It is
717 somewhat reckless for us to exclude it in our implementation.

## J  What Happens to the Representations during Learning

During training, the gradient descent tries to find representations of the puzzle that require less and less information to encode. This information is measured by the KL term for $z$, plus the a heavily penalized reconstruction error.

Due to the 10x penalization on reconstruction error, and the initial high capacity for $z$, the $z$ distribution (which we call the "posterior") quickly learns the information that is required to perfectly reconstruct the given input/output pairs in the puzzle, within the first 20 or so steps. The remainder of the training steps are about compressing $z$ information under the constraint of perfect reconstruction, by tuning the representations to be more concise.

Our mental model of how gradient descent compresses the $z$ information consists of several steps which we list below:

1. Suppose the posterior $p$ originally codes for some number $n$ pieces of information $z_1, \ldots, z_n$ using thin Gaussians.

2. The posterior widens and becomes more noisy to try to get closer to the wide Gaussian "prior" $q = N(0, 1)$, but since all $n$ pieces of information are needed to ensure good reconstruction, the noise is limited by the reconstruction loss incurred.

3. The ever-widening posteriors push the neurons to become more and more resilient to noise, until some limit is reached.

4. Learning remains stagnant for a while, as a stalemate between compression and reconstruction.

5. If it turns out that $z_1$ is not reconstructible using $z_2, \ldots, z_n$, then stop. Else, proceed to step 6.

6. The neurons, pushed by the widening posterior of $z_1$, figure out a procedure to denoise $z_1$ using information from $z_2, \ldots, z_n$, in the event that the noise sample for $z_1$ is too extreme.

7. The posterior for the last piece keeps pushing wider, producing more extreme values for $z_1$, and the denoising procedure is improved, until the $z_1$ representation consists completely of noise, and its usage in the network is replaced by the output of the denoising procedure.

8. The posterior for $z_1$ is now identical to the prior, so nothing is coded in $z_1$ and it no longer contributes to the KL loss.

9. The posterior now codes for $n-1$ pieces of information $z_2, \ldots, z_n$, and compression has occurred.

These steps happen repeatedly for different unnecessarily coded pieces of information, until there are no more. More than one piece of information can be compressed away at once, and there is no need for the steps to proceed serially. The process stops when all information coded by the posterior is unique, and no piece is reconstructable using the others.

## K  Additional Details about the ARC-AGI Benchmark

Figure 12 shows three examples of ARC-AGI-1 training puzzles.

For every puzzle, there is a hidden rule that maps each input grid to each output grid. You are given some number of examples of input-to-output mappings, and you get **two attempts** to guess the output grid for a given input grid, without being told the hidden rule. If either guess is correct, then you score 1 for that puzzle, else you score 0. Some puzzles have more than one input/output pair that you have to guess, in which case the score for that puzzle may be in between.

The main ideas that the training puzzles aim to teach can be described more elaborately as follows:

- **Objectness**: Objects persist and cannot appear or disappear without reason. Objects can interact or not depending on the circumstances.

- **Goal-directedness**: Objects can be animate or inanimate. Some objects are "agents" - they have intentions and they pursue goals.

- **Numbers & counting**: Objects can be counted or sorted by their shape, appearance, or movement using basic mathematics like addition, subtraction, and comparison.

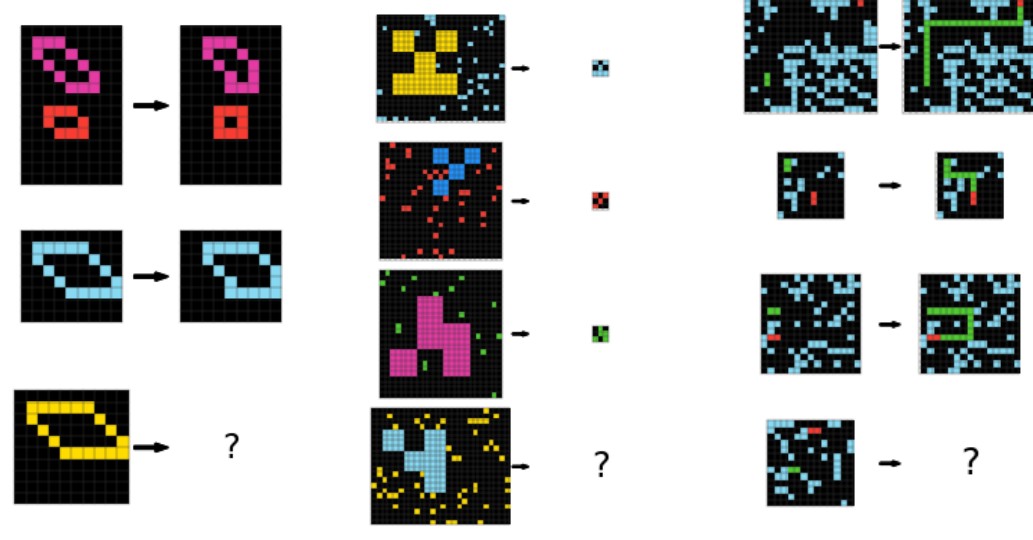

(a) **Hidden rule:** Shift every object right by one pixel, except the bottom/right edges of the object.

(b) **Hidden rule:** Shrink the big object and set its color to the scattered dots' color.

(c) **Hidden rule:** Extend the green line to meet the red line by turning when hitting a wall.

Figure 12: Three example ARC-AGI-1 puzzles.

- **Basic geometry & topology**: Objects can be shapes like rectangles, triangles, and circles which can be mirrored, rotated, translated, deformed, combined, repeated, etc. Differences in distances can be detected.

The competitions launched by the ARC Prize Foundation have been restricted to 12 hours of compute per solution submission, in a constrained environment with no internet access. This is where a hidden semi-private evaluation set is used to score solutions. The scores we report are on the public evaluation set, which is of the same difficulty as the semi-private evaluation set, which we had no access to when we performed this work. The scores we listed for reasoning models were achieved with compute budgets well over the limits of the constrained environment. Otherwise, all other solutions we mention are scored on the semi-private evaluation set within the competition constraints.

## L    Additional Case Studies

Below, we show two additional puzzles and a dissection of CompressARC's solution to them.

### L.1    Case Study: Bounding Box

Puzzle 6d75e8bb is part of the training split, see Figure 13.

#### L.1.1    Watching the Network Learn: Bounding Box

**Human Solution:** We first realize that the input is red and black, and the output is also red and black, but some of the black pixels are replaced by light blue pixels. We see that the red shape remains unaffected. We notice that the light blue box surrounds the red shape, and finally that it is the smallest possible surrounding box that contains the red shape. At this point, we copy the input over to the answer grid, then we figure out the horizontal and vertical extent of the red shape, and color all of the non-red pixels within that extent as light blue.

**CompressARC Solution: See Table 5**

#### L.1.2    Solution Analysis: Bounding Box

Figure 14 shows the amount of contained information in every tensor composing the latent $z$.

Table 5: CompressARC learning the solution for Bounding Box, over time.

| Learning steps | What is CompressARC doing? | Sampled solution guess |
|---|---|---|
| 50 | The average of sampled outputs shows that light blue pixels in the input are generally preserved in the output. However, black pixels in the input are haphazardly and randomly colored light blue and red. CompressARC does not seem to know that the colored input/output pixels lie within some kind of bounding box, or that the bounding box is the same for the input and output grids. | sample / sample average / guess 1 / guess 2 |
| 100 | The average of sampled outputs shows red pixels confined to an imaginary rectangle surrounding the light blue pixels. CompressARC seems to have perceived that other examples use a common bounding box for the input and output pixels, but is not completely sure about where the boundary lies and what colors go inside the box in the output. Nevertheless, guess 2 (the second most frequently sampled output) shows that the correct answer is already being sampled quite often now. | sample / sample average / guess 1 / guess 2 |
| 150 | The average of sampled outputs shows almost all of the pixels in the imaginary bounding box to be colored red. CompressARC has figured out the answer, and further training only refines the answer. | sample / sample average / guess 1 / guess 2 |

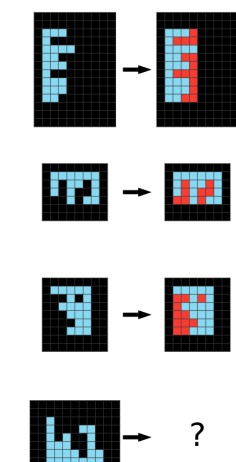

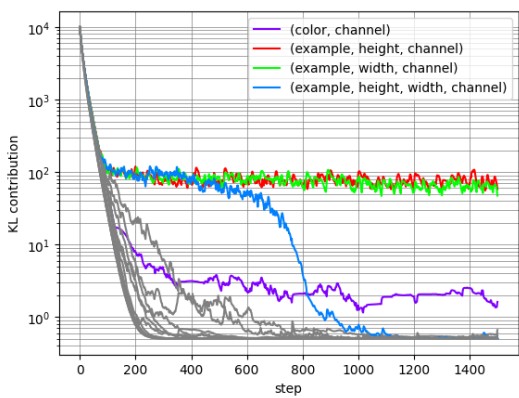

Figure 13: Bounding Box: Puzzle 6d75e8bb from the training split.

Figure 14: Breaking down the KL loss during training into contributions from each individual shaped tensor in the multitensor $z$.

All the tensors in $z$ fall to zero information content during training, except for three tensors. From 600-1000 steps, we see the (example, height, width, channel) tensor suffer a massive drop in information content, with no change in the outputted answer. We believe it was being used to identify the light blue pixels in the input, but this information then got memorized by the nonlinear portions of the network, using the (example, height, channel) and (example, width, channel) as positional encodings.

Figure 15 shows the average output of the decoding layer for these tensors to see what information is stored there.

### L.2   Case Study: Center Cross

Puzzle 41e4d17e is part of the training split, see Figure 16a.

**Human Solution:** We first notice that the input consists of blue "bubble" shapes (really they are just squares, but the fact that they're blue reminds us of bubbles) on a light blue background and the output has the same. But in the output, there are now magenta rays emanating from the center of each bubble. We copy the input over to the answer grid, and then draw magenta rays starting from the center of each bubble out to the edge in every cardinal direction. At this point, we submit our answer and find that it is wrong, and we notice that in the given demonstrations, the blue bubble color is drawn on top of the magenta rays, and we have drawn the rays on top of the bubbles instead. So, we pick up the blue color and correct each point where a ray pierces a bubble, back to blue.

**CompressARC Solution:** We don't show CompressARC's solution evolving over time because we think it is uninteresting; instead will describe. We don't see much change in CompressARC's

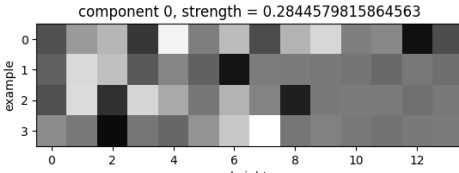

(a) (**example**, **height**, **channel**) **tensor.** The first principal component is 771 times stronger than the second principal component. **A brighter pixel indicates a row with more light blue pixels.** It is unclear how CompressARC knows where the borders of the bounding box are.

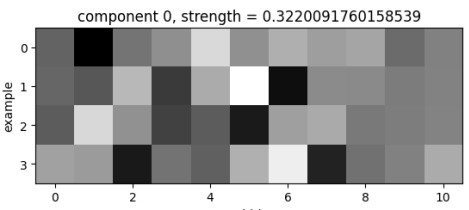

(b) (**example**, **width**, **channel**) **tensor.** The first principal component is 550 times stronger than the second principal component. **A darker pixel indicates a column with more light blue pixels.** It is unclear how CompressARC knows where the borders of the bounding box are.

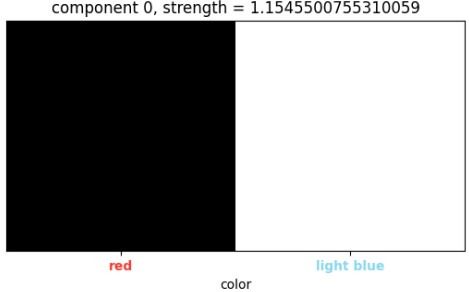

(c) (**color**, **channel**) **tensor.** This tensor serves to distinguish the roles of the two colors apart.

Figure 15: Breaking down the loss components during training tells us where and how CompressARC prefers to store information relevant to solving a puzzle.

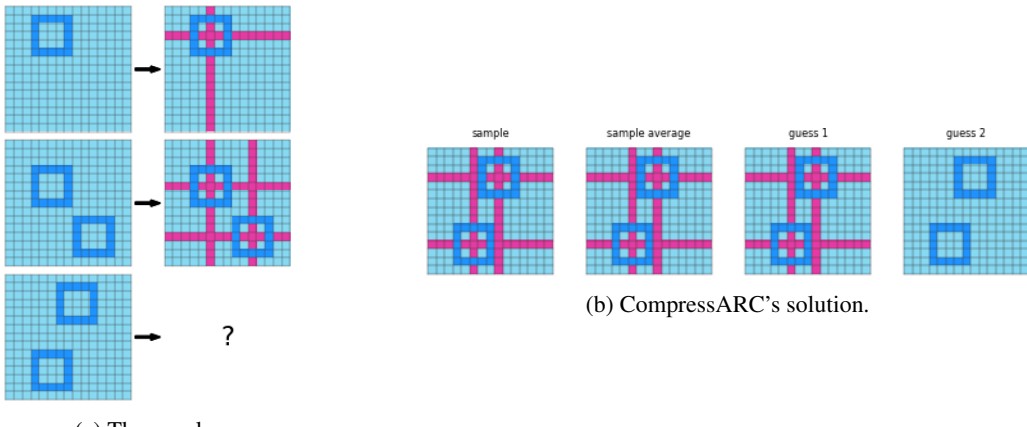

(a) The puzzle.

(b) CompressARC's solution.

Figure 16: Center Cross: Puzzle 41e4d17e from the training split.

answer over time during learning. It starts by copying over the input grid, and at some point, magenta rows and columns start to appear, and they slowly settle on the correct positions. At no point does CompressARC mistakenly draw the rays on top of the bubbles; it has always had the order correct.

### L.2.1 Solution Analysis: Center Cross

Figure 17 shows another plot of the amount of information in every tensor in $z$. The only surviving tensors are the $(\text{color}, \text{channel})$ and $(\text{example}, \text{height}, \text{width}, \text{channel})$ tensors, which are interpreted in Figure 18.

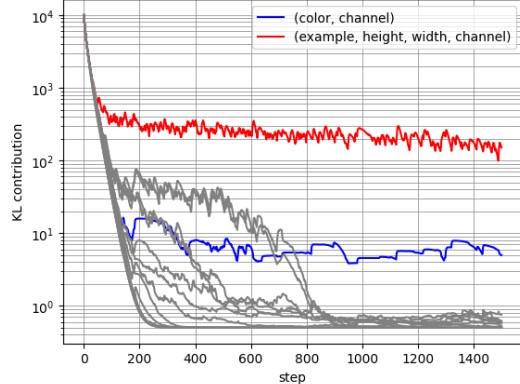

Figure 17: Breaking down the KL loss during training into contributions from each individual shaped tensor in the multitensor $z$.

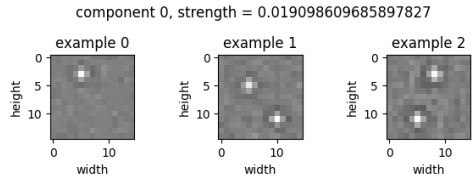

(a) $(\textbf{example}, \textbf{height}, \textbf{width}, \textbf{channel})$ **tensor.** The top principal component is 2496 times stronger than the second principal component. **This tensor codes for the centers of the bubbles.** In the KL contribution plot, we can see that the information content of this tensor is decreasing over time. Likely, CompressARC is in the process of eliminating the plus shaped representation, and replacing it with a pixel instead, which takes fewer bits.

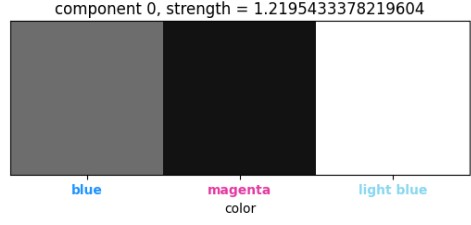

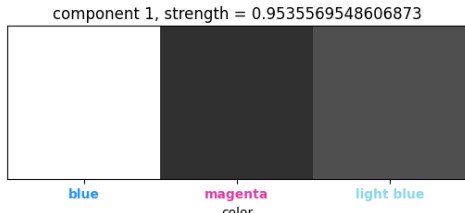

(b) $(\textbf{color}, \textbf{channel})$ **tensor.** This tensor just serves to distinguish the individual roles of the colors in the puzzle.

Figure 18: Breaking down the loss components during training tells us where and how CompressARC prefers to store information relevant to solving a puzzle.

## M   List of Mentioned ARC-AGI-1 Puzzles

See Table 6 below.

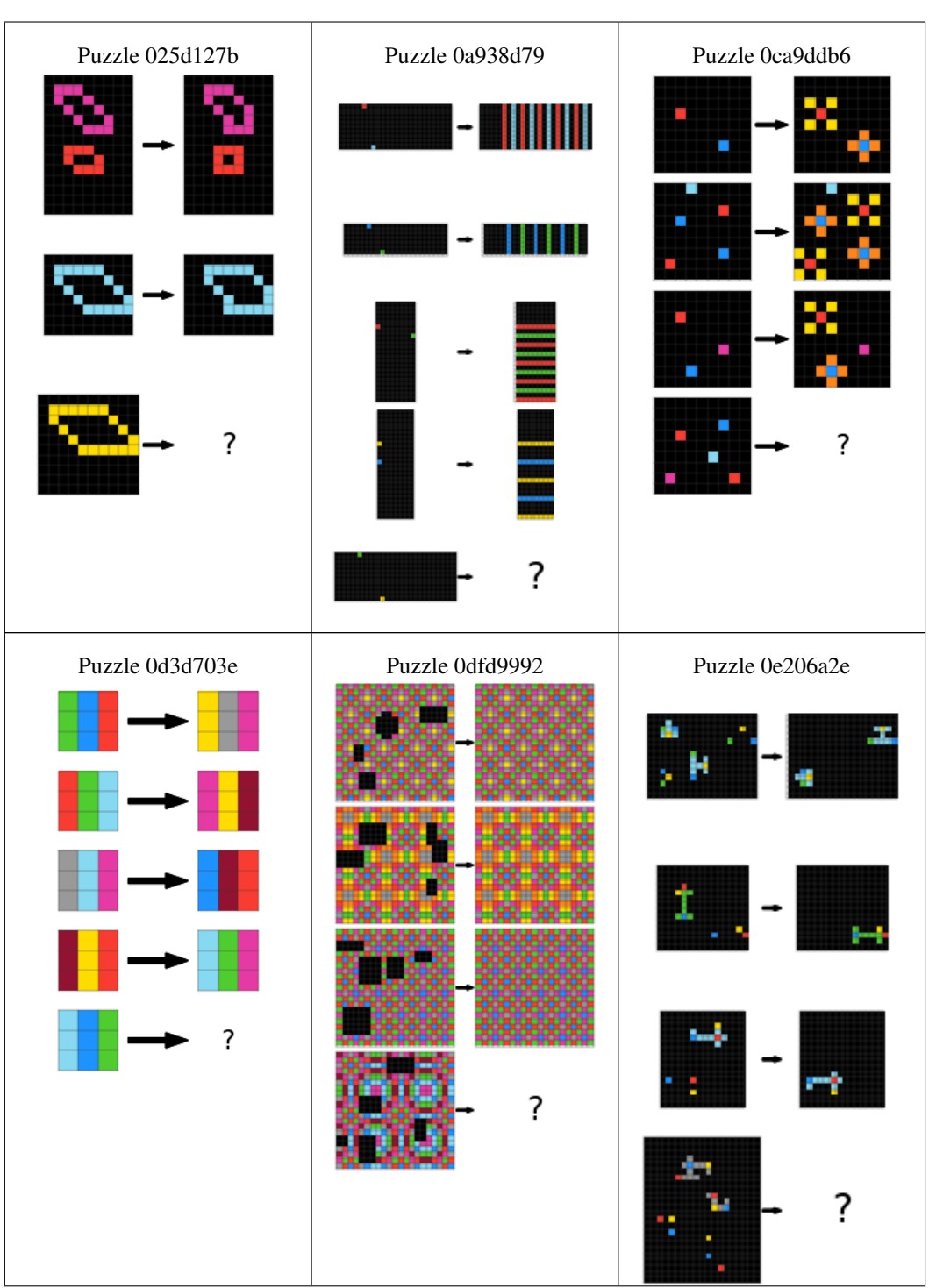

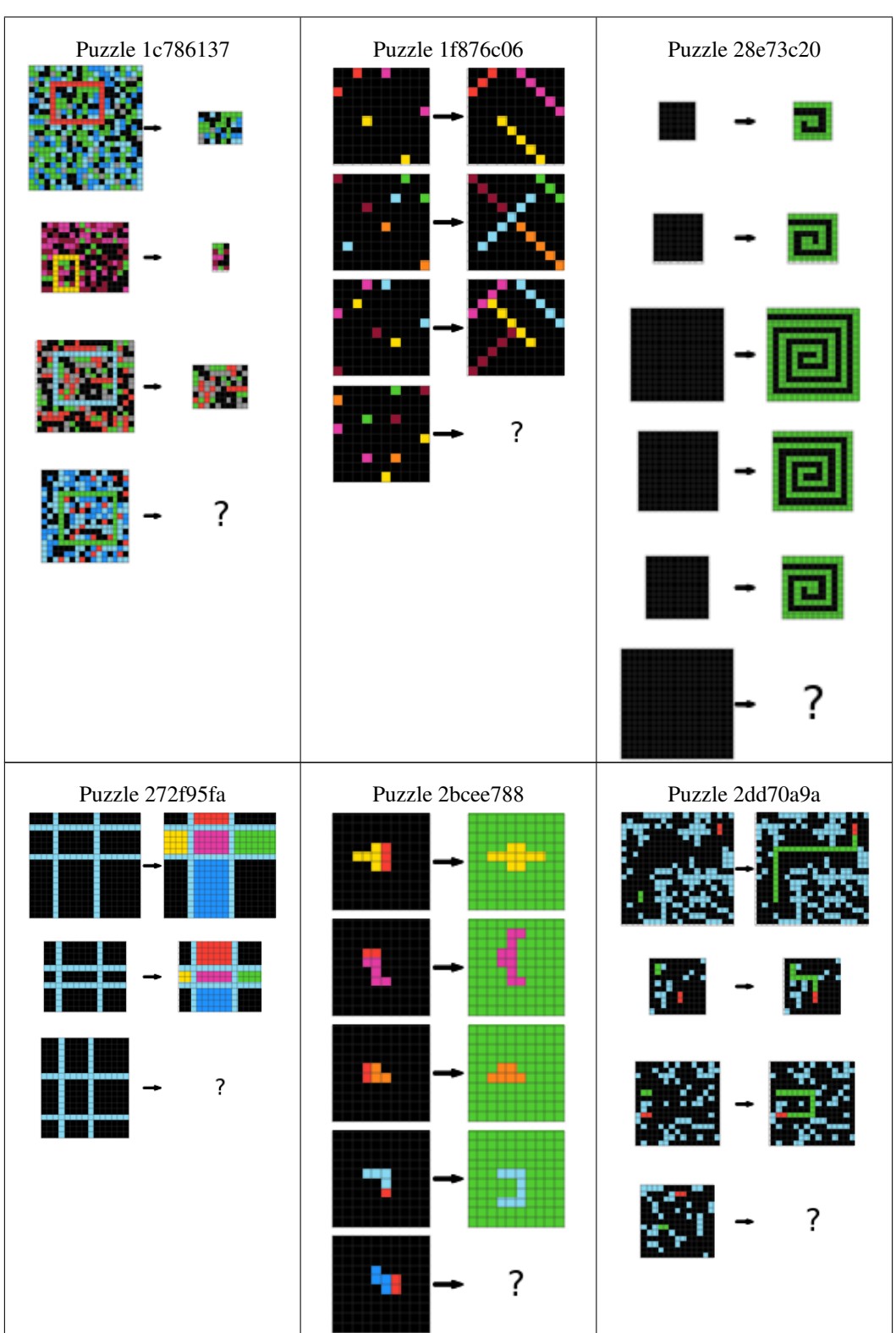

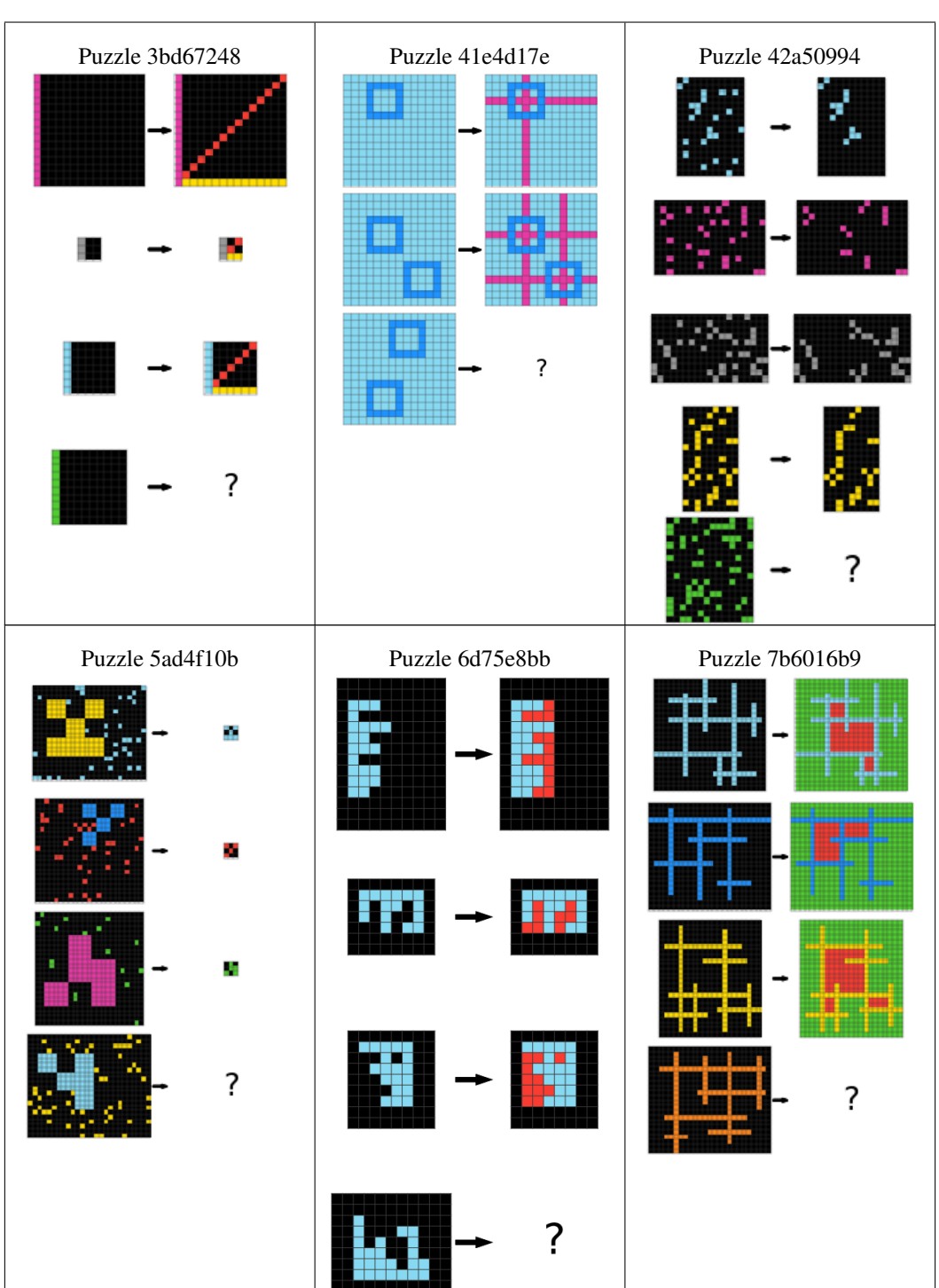

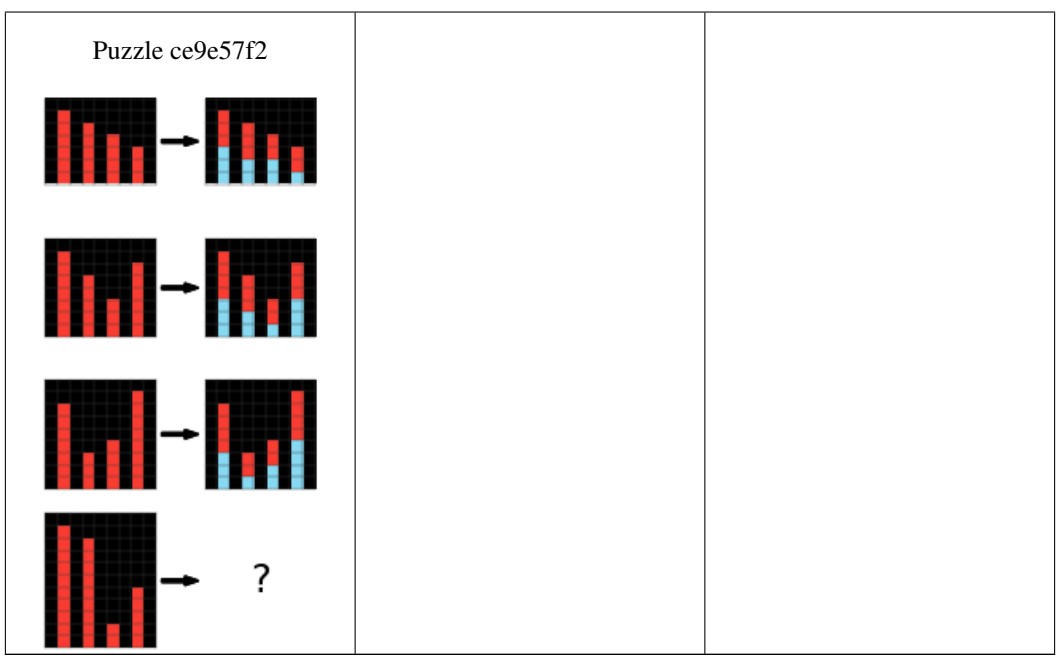

Table 6: List of Mentioned ARC-AGI=1 Puzzles. All these puzzles are part of the training split.

## N  Code

Code for this project is provided in the supplemental materials.

