# OpenReview forum: "ARC-AGI Without Pretraining"
_NeurIPS.cc/2025/Conference — Submitted to NeurIPS 2025_

### Official Review · Reviewer_2pAi · 2025-07-02

**Clarity:** 3
**Significance:** 4
**Originality:** 4
**Rating:** 4
**Confidence:** 3

**Summary:**

The paper presents a test-time learning approach that can solve 20% of the evaluation puzzles from the ARC-AGI benchmark. The approach involves minimizing the minimum description length of the target puzzle at inference time. Further, the approach is demonstrated to work with a model that is not pre-trained. While the approach does not produce SOTA performance on ARC-AGI benchmark it is still interesting in that it demonstrates the strength of unsupervised learning at inference time. The paper analyzes a number of cases where the approach works and does not work as expected.

**Questions:**

- The experiments could be strengthened if an appropriate baseline could be added. Specifically, it would be useful to see how a network that is trained on the test-time puzzle would work on the ARC benchmark.

- The paper and contribution could be strengthened further if evaluated on another domain where the test-time performance is crucial. Are there other domains where the authors think it would be useful to evaluate their approach? And would it be possible to add them into the main text or in the appendix?

- The presented can benefit from presenting the loss functions and the pseudocode related to the approach in the main text.

- The approach involves minimizing the MDL on the test time puzzle. The paper does not seem to discuss or experiment with other unsupervised learning approaches (reconstruction objectives).

- The experiment results seem to be coming from a single training seed. It would be useful to understand whether the approach can produce similar results when trained with different random seeds.

I would be happy to increase the scores if baselines and other unsupervised approaches are compared in the experiments.

**Ethical Concerns:**

["NO or VERY MINOR ethics concerns only"]

**Final Justification:**

I have read the rebuttal and it has addressed some of my questions I raised during my review. I would stick to original rating of "4".

**Limitations:**

Yes, the authors adequately addressed the limitations and potential negative societal impact of their work.

**Paper Formatting Concerns:**

N / A

**Quality:**

3

**Strengths And Weaknesses:**

Strengths:
- The paper is clearly written and the approach is well presented.
- The contribution and the work are significant.
- The approach tackles an important problem of test-time learning which is becoming more and more relevant in language model research.
- Extensively studies how the unsupervised training approach works on the ARC-AGI benchmark by providing cases and scenarios. Also, presents failure cases where the approach cannot solve the given task.

Weaknesses:
- The approach is validated on one benchmark and that brings out the question of whether the approach and insights can transfer / scale to more domains.
- Does not present the pseudocode and loss functions relevant to the approach. These would be useful for reproducing the results from the paper.

---

> ### Author Rebuttal · Authors · 2025-07-31
>
> We thank the Reviewer for their thoughtful evaluation of our work, and their comments which help us to improve our work and it's presentation.
>
> **Addressing weakness #1:** See below for rebuttal to question 2.
>
> **On weakness #2:**
>
> Multiple reviewers have pointed this out, and we agree with this assessment. To improve the clarity of our method as written, we will choose to include three algorithm/pseudocode blocks in Section 3 in the camera ready version. See our rebuttal to Reviewer DdUE for these algorithms. Algorithms 2 and 3 will contain the relevant loss functions. For reproducibility, the code for reproducing the results is already in the supplimentary material. This contains full instructions for usage as well as tips to help you read the code.
>
> Otherwise, we completely agree with all the rest of the strengths and weaknesses listed.
>
> **For the questions:**
> 1. We will also choose to add a table with baselines/comparisons to Section 5. This table is shown in our rebuttal to Reviewer rsDF. It will include the method "Test-Time Training" [1], which is pre-trained and also trains on the test-time puzzle. We do not know of other results that train exclusively on the test-time puzzle, if that is what you are looking for. Due to timing constraints of the rebuttal period, we are unable to run/show any experiments to compare against direct supervised learning during test time.
> 1. This work is mostly a proof of concept for the MDL approach towards intelligence. Most of the methodology is not transferrable to other downstream tasks. The main part that is transferrable, is the idea of how to transform the MDL objective into a problem of gradient descent during inference time, as explained in Section 3. This is because the statements of Section 3 can still apply when we replace:
>     - "solving ARC-AGI puzzles" -> "infilling partially observed data"
>     - "incomplete puzzle" -> "observed data"
>     - "answers" -> "unobserved data"
>
>     Looking at our method as a data infilling methodology may help other researchers make similar derivations of more new methods for other domains like supervised learning and generative modeling.
>
>     While we cannot add any empirical results on any such method (our architecture is ARC-AGI-specific,) we can posit this extension as a theoretical possibility in the camera-ready version of the paper if the Reviewer would like. That said, we feel that this kind of suggestion may be too speculative and unsubstantiated to include in published work, and we would otherwise like to avoid it.
>
> 1. See our address to weakness #2.
> 1. We are unsure which other reconstruction objectives the Reviewer wants us to discuss/experiment with. Two possibilities that we can cover:
>     - Other unsupervised learning approaches known to the literature. Continuing discussion from the Reviewer's question 1, the new baselines/comparisons table will also include the method "Latent Program Network" [2]. Latent Program Network includes a VAE setup, which is an unsupervised learning approach with a reconstruction objective. This method is also mentioned in the second bullet point in Appendix H, but we will now also mention it's unsupervised objective in the table as well.
>     - The Reviewer wants us to use a reconstruction objective ourselves. CompressARC does indeed use a reconstruction objective, see crossentropy in Algorithm 2 from our rebuttal to Reviewer DdUE. The loss we compute in Algorithm 3 has two components, one of which is the reconstruction objective. We apologize for any confusion caused by our original description of this.
> 1. We are unable to get this experiment set up in time, but we agree this is a good idea, and will report variance in the next version of this paper. To try to alleviate the Reviewer's concerns, we remind the Reviewer by introducing Algorithm 3 that an entirely separate architecture is trained for every puzzle, and a different quantity of parameters are initialized and learned in each case. Since the initialization must have changed for every puzzle, there is already a degree of per-puzzle random variation involved that reduces the ways in which bias can enter our experiments.
>
> We thank the Reviewer once again for the discussion, and we hope we have addressed their questions satisfactorily. We invite the Reviewer to discuss further on any aspect of our work, if they so desire.
>
> **References:**
>
> [1] Akyürek, Ekin, et al. "The surprising effectiveness of test-time training for few-shot learning." Forty-second International Conference on Machine Learning, 2024.
>
> [2] Macfarlane, Matthew V., and Clément Bonnet. "Searching latent program spaces." arXiv preprint arXiv:2411.08706 (2024).

---

### Official Review · Reviewer_rsDF · 2025-07-02

**Clarity:** 1
**Significance:** 3
**Originality:** 3
**Rating:** 4
**Confidence:** 3

**Summary:**

The paper introduces CompressARC, a novel model designed to solve ARC-AGI-1 benchmark puzzles without any pretraining. It uses Minimum Description Length (MDL) to solve puzzles directly at inference time. It achieves a 20% success rate on evaluation puzzles, demonstrating surprising generalization given its extremely data-limited setting—using only the incomplete test puzzle itself.

**Questions:**

See Strengths and Weaknesses.

**Ethical Concerns:**

["NO or VERY MINOR ethics concerns only"]

**Final Justification:**

Following the rebuttal, Weaknesses 2 and 3 are satisfactorily addressed—the relationship between the reconstruction loss and MDL is now clearer, and the added baselines (including TTT) strengthen the empirical case. However, improvements in presentation and clarity are still needed, particularly in balancing the level of methodological abstraction with domain-specific implementation details. Therefore, I'd like to keep my original score unchanged.

**Limitations:**

See Strengths and Weaknesses.

**Quality:**

3

**Strengths And Weaknesses:**

**Strengths:**

1. The core idea and task formulation presented in this work are both innovative and well-motivated. The use of MDL (Minimum Description Length) to guide inference-time learning is particularly novel and conceptually elegant.
2. The empirical results are impressive and substantiate the effectiveness of MDL in tackling the ARC-AGI benchmark. The demonstrated generalization performance under minimal data and training constraints is especially compelling.

**Weaknesses:**

1. **Presentation and Clarity:**
   The paper suffers from presentation issues that make it difficult to follow in key sections.

   * Lines 110–123 in Section 3 would benefit greatly from being reformulated as an explicit algorithm or pseudocode subsection to enhance clarity.
   * Section 4, which describes the architecture, is underdeveloped. Figure 2 is difficult to interpret, and the caption does little to clarify its content. More detail should be provided on the model components and their interaction.
   * The paper would benefit from allocating more space to Sections 3 and 4, which are central to understanding the methodology, rather than dedicating extensive discussion to Section 2.


2. **Unclear Learning Objective:**
   While the paper claims to leverage MDL principles, the actual optimization appears to minimize a reconstruction loss. This raises an important question: how exactly does the reconstruction loss relate to MDL in this context? The connection between the two should be made more explicit—ideally with a theoretical or empirical justification. As it stands, the objective being optimized does not clearly reflect MDL, which could undermine the central premise of the approach.


3. **Missing Baselines:**
   The paper lacks comparisons to relevant baselines, particularly *Test-Time Training (TTT)* \[1], which is closely related in spirit to the proposed method. Including such baselines would strengthen the empirical claims and better contextualize the contribution.



**Reference:**

\[1] Akyürek, Ekin, et al. "The surprising effectiveness of test-time training for few-shot learning." *Forty-second International Conference on Machine Learning*, 2024.

---

> ### Author Rebuttal · Authors · 2025-07-31
>
> We thank the Reviewer for their thoughtful review and their suggestions on how to improve the clarity of our paper.
>
> **On strengths:** We thank the Reviewer for recognizing the generalization ability of our highly data-limited method.
>
> **On presentation and clarity:**
>
> - Thank you for the suggestion, we agree that an algorithm section would improve the clarity. We will replace lines 110-123 with a new section, see Algorithm 2 in our rebuttal for Reviewer DdUE. For consistency, we will also change lines 126-129 to an algorithm section as well. (See Algorithm 1 in our rebuttal for Reviewer DdUE.)
> - Seeing that the architecture is the part of the paper that is the least insightful/transferrable to other domains, we decided to only give a brief overview in the main text, and include a fully detailed description of the architecture's design in Appendices B and C for more enthusiastic readers and for completeness.
> - We would like to emphasize in Section 4 that the architecture is heavily engineered to be used specifically for ARC-AGI, and that the engineering details are less transferrable to other domains, making them less important to include than, say, the broader idea of making the architecture respect lots of symmetries. The reader learns less from these details, so they are instead included in Appendices B and C for completeness and for more eager readers. If we would like to keep Section 4 brief for this reason, may we offer to the Reviewer to replace the sentence on line 149-150 with the following?
>     > We warn the reader that the model components are heavily engineered and very domain-specific. This makes them less informative on the overall idea behind CompressARC's approach, which merely requires that the architecture as a whole respects the symmetries of the problem domain. As such, we merely list the architecture components here, and describe each one in full detail in the Appendix:
>
>     We agree that Section 2 does not need its current amount of detail. In the camera ready version, we will move most of it's detail to Appendix K, to make space for the new algorithm sections.
>
> **On the unclear learning objective:**
>
> We apologize for the confusion here. We hope the suggested algorithm sections clear this up. Algorithm 1 serves as the literal description in MDL, and Algorithm 2 minimizes its length. Algorithm 2 produces the puzzle's solution as an intermediate value, so Algorithm 3, which is CompressARC (see Algorithm 3 in our rebuttal for Reviewer DdUE,) is just a truncated version of Algorithm 2 that only outputs the solution. The reconstruction term in the loss calculates the expected length of line 3 of Algorithm 1, and the KL term in the loss calculates the expected length of line 1. This is where the connection between the loss function and the MDL comes from.
>
> **On the missing baselines:**
>
> Thank you for the suggestion, we agree that adding baselines will better contextualize and strengthen the claims in our paper. We will include the following information in a new table in Section 5:
>
> | Method                              | Reference      | Score     | Training data incorporated     | Similarity to ours |
> | --- | --- | --- | --- | --- |
> | o3, high compute                    | [1]            | 87.5%     | LLM training data              | flexible inference compute |
> | Test-time Training                  | [2]            | 53.0%     | LLM training data              | training during inference time |
> | LLM-powered program synthesis       | [3]            | 42.0%     | LLM training data              | - |
> | GPT-4o, direct prompting (pass@1)   | [3]            |  9.0%     | LLM training data              | - |
> | Hierarchical Reasoning Model        | [4]            | 40.3%     | only ARC-AGI puzzles           | flexible inference compute |
> | Latent Program Network              | [5]            |  3.0%     | only ARC-AGI puzzles           | VAE-like unsupervised loss function |
> | Brute force search over DSL         | [3]            | 20.0%     | None                           | No training data |
> | **CompressARC (ours)**                  |  -             | 20.0%     | None                           | - |
>
> These numbers obtained directly from the cited papers. We will move Table 1 to the Appendix to make space for this new table.
>
> We thank the Reviewer once again for the discussion, and we hope we have addressed their questions satisfactorily. We invite the Reviewer to discuss further on any aspect of our work, if they so desire.
>
> **References:**
>
> [1] Chollet, F. "OpenAI o3 Breakthrough High Score on ARC-AGI-Pub." ARC Prize, 2024.
>
> [2] Akyürek, Ekin, et al. "The surprising effectiveness of test-time training for few-shot learning." Forty-second International Conference on Machine Learning, 2024.
>
> [3] Chollet, F., Knoop, M., Kamradt, G., & Landers, B. (2024). Arc prize 2024: Technical report. arXiv preprint arXiv:2412.04604.
>
> [4] Wang, G., Li, J., Sun, Y., Chen, X., Liu, C., Wu, Y., ... & Yadkori, Y. A. (2025). Hierarchical Reasoning Model. arXiv preprint arXiv:2506.21734.
>
> [5] Macfarlane, Matthew V., and Clément Bonnet. "Searching latent program spaces." arXiv preprint arXiv:2411.08706 (2024).

---

> > ### Comment · Reviewer_rsDF · 2025-08-05
> >
> > Thank you for explanations and including additional experiments. My concerns about weakenss 2 and 3 are resolved, while weakness 1 remains unresolved because the revised Algorithms 1/2 remain difficult to follow. What's "Red text" in algorithm 1?

---

> > > ### Author Response · Authors · 2025-08-05
> > >
> > > We apologize for the note on red text; it is a typo left when we were refining the table format to show you here. What is meant is, in the final version, text in Algorithm 1 that is within angle brackets (<>) will also be colored red in order to highlight that it is meant to be filled in with a hardcoded value picked by Algorithm 2.

---

### Official Review · Reviewer_DdUE · 2025-07-02

**Clarity:** 1
**Significance:** 3
**Originality:** 3
**Rating:** 4
**Confidence:** 3

**Summary:**

The paper approaches the ARC-AGI benchmark as one of finding a small program to explain the relationship between the input and the output examples. This is done by training on the examples themselves and, thus, does not require any pre-training. The method learns the mean and co-variance matrix of a normal distribution, samples from this and gives this as input to a neural network which outputs the input/output pairs in the problem. The neural network is trained to reconstruct the input/output pairs and KL divergence is used to make the noise distribution as close to a standard normal as possible.

The paper shows that more training time results in a greater percentage of puzzles being solved. While training required to solve 20% of test instances requires 138 hours, no pre-training is required.

**Questions:**

In Figure 1, what are the different layers shown in the left two plots?
Where does seed_error come from?

**Ethical Concerns:**

["NO or VERY MINOR ethics concerns only"]

**Final Justification:**

Given the additional algorithmic clarity, I am adjusting my score from a 3 to a 4.

**Limitations:**

The results appear to show that training is done at about a rate of 15 iterations per hour. Is this for a single example or across all examples? Even so, 15 iterations per hour seems slow. Why is this?

**Quality:**

2

**Strengths And Weaknesses:**

**Strengths**
The paper shows that ARC-AGI puzzles can be solved without any pre-training. The only data needed is the data from the puzzle.

**Weaknesses**
The methods in the paper are not clearly presented and many details are buried in the appendix. It is unclear how the parameters of the normal distribution are learned. There is pseudocode given starting at line 126 that shows a seed_z and seed_error. It says these are picked by relative entropy coding (REC). However, this appears to conflict with the overall method of learning a mean and co-variance matrix for a Normal distribution. In fact, this is the only time seed_error is mentioned in the main paper.

The paper also does not contextualize the results in the broader research landscape. For someome unfamiliar with the ARC-AGI benchmark, one cannot know how these results compare to other existing results, especially for those that do not rely on pre-training. Is this method the only one that does not rely on pretraining?

---

> ### Author Rebuttal · Authors · 2025-07-31
>
> We thank the Reviewer for constructive criticism and thoughtful questions that help us improve the clarity of our paper.
>
> Multiple reviewers had concerns about clarifying where the training objective comes from, what is learned, and what happens to the seeds. As such, we decided that if our paper is accepted, **our camera ready version will include three Algorithm boxes** in Section 3 to better explain our method:
>
> ```
> Algorithm 1: Short program that outputs puzzle P with any solution S. Red text is to be substituted in with hard-coded values produced via Algorithm 2.
> ---
> Set seed_z = <seed_z>;                                                      Hardcoded info about z
> Set theta = <theta>;                                                        Hardcoded weights
> Set seed_error = <seed_error>;                                              Hardcoded prediction error info
> z <- sample(N(0,1), seed_z);                                                Generate inputs z
> (puzzle_logits, solution_logits) <- <equivariant_neural_net>(z, theta);     Forward pass
> P <- sample(Categorical(puzzle_logits), seed_error);                        Generate puzzle
> S <- sample(Categorical(solution_logits), seed_error);                      Generate solution
> Return P, S
> ```
>
>
> ```
> Algorithm 2: Minimize Description Length.
> ---
> Input: Puzzle P;
> Pick equivariant_neural_net architecture dimensions that match up with the data shape of P;
> Initialize mu, diagonal Sigma, and weights theta;
> foreach step do
>     z <- sample(N(mu, Sigma));
>     (puzzle_logits, solution_logits) <- equivariant_neural_net(z, theta);
>     L <- KL(N(mu, Sigma)||N(0,1)) + crossentropy(puzzle_logits, P);            approximately equal to len(Algo 1) + C
>     mu, Sigma, theta <- Adam(grad_mu L, grad_Sigma L, grad_theta L);
> end foreach
> seed_z <- REC(N(mu, Sigma), N(0,1));                                           Store z with KL(N(mu, Sigma)||N(0,1)) bits
> z <- sample(N(0,1), seed_z);
> (puzzle_logits, _) <- equivariant_neural_net(z, theta);
> seed_error <- REC(Categorical(puzzle_logits), delta_P);                        Store puzzle with crossentropy bits
> description <- code for Algorithm 1, with values of seed_z, theta, seed_error,
>                and the function equivariant_neural_net substituted in;         Write down Algo 1
> Return description
> ```
> Note that the KL term in the loss measures the length of line 1 in Algorithm 1, and the reconstruction term measures the length of line 3.
>
> ```
> Algorithm 3: CompressARC. Same as Algorithm 2, but truncated to only return solution_logits.
> ---
> Input: Puzzle P;
> Pick equivariant_neural_net architecture dimensions that match up with the data shape of P;
> Initialize mu, diagonal Sigma, and weights theta;
> foreach step do
>     z <- sample(N(mu, Sigma));
>     (puzzle_logits, solution_logits) <- equivariant_neural_net(z, theta);
>     L <- KL(N(mu, Sigma)||N(0,1)) + crossentropy(puzzle_logits, P);            approximately equal to len(Algo 1) + C
>     mu, Sigma, theta <- Adam(grad_mu L, grad_Sigma L, grad_theta L);
> end foreach
> Return solution_logits
> ```
>
> We will move some of Section 2 to the Appendix to create the space for this.
>
> **Clarifying the parts of the method mentioned in weaknesses:**
>
> Algorithm 1 above acts as our small description of the puzzle and Algorithm 2 is our proposed way of generating the hardcoded values for Algorithm 1. Algorithm 2 minimizes the description length, and it creates the seed_z and seed_error to hard-code into Algorithm 1. Algorithm 2 also generates puzzle solutions in the middle during it's run as a side effect. So to solve ARC-AGI puzzles, we only need to run it up to that point, which is how we get Algorithm 3, the one we actually run, which learns a mean and covariance together alongside the weights. Algorithm 3 skips the steps of Algorithm 2 that compute seeds, since this is unnecessary to obtain the puzzle solution. This is why seed_error never appears in the implementation.
>
> **On the question asked:**
>
> In Figure 1, the second-from-the-left boxes represent the input to the equivariant neural network; they are learned tensors just like the weights theta, rather than layers. Even though they get fed as input into the layers, they are also learned parameters alongside the parameters of the layers themselves. The leftmost boxes represent a standard normal distribution. They show that the inputs contribute a loss as we learn them, and that loss is a KL divergence to the standard normal distribution.
>
> **On the limitation:**
>
> This is 15 iterations per hour if training for all 400 puzzles at once. This is slow because we have been solving the puzzles one at a time, and the GPU is very under-utilized. Due to timing constraints of the rebuttal period, we have so far been unable to produce a working parallelized version to gauge how much faster we might be able to go.
>
> We thank the Reviewer once again for the discussion, and we hope we have addressed their questions satisfactorily. We invite the Reviewer to discuss further on any aspect of our work, if they so desire.

---

> > ### Comment · Reviewer_DdUE · 2025-08-09
> >
> > Thank you, this clarifies the question I had about seed_z and seed_error.

---

### Official Review · Reviewer_umP5 · 2025-07-06

**Clarity:** 3
**Significance:** 2
**Originality:** 4
**Rating:** 4
**Confidence:** 4

**Summary:**

Introduces a minimum description length approach/model capable of solving a significant subset of ARC-AGI-1 evaluation puzzles without pre-training. Their method trains a neural net on individual puzzles to predict an output grid from a parameterized latent/learned noise and suggests an pre-training alternative to modeling intelligence. Key advantages include lower computation requirements from foregoing pretraining.

**Questions:**

Questions and Suggestions on 3-5 key points
1. Does only including Z in the “description length” proxy underestimate the efficiency/compactness of the learned program by not including the “decoding” program given by the neural net? I could be misunderstanding but it certainly seems that the neural net is deeply load bearing for learning the transform.
2. Is this methodology easily transferable to other downstream tasks? MDL approaches on the whole are definitely more generalizable, but are there key takeaways/innovations from this work that enables new use cases? Perhaps this is more of a proof of concept for MDL approach on something abstract like ARC’s spatial reasoning task.
3. Are there early results available for the joint-compression/cross-puzzle weight-sharing ideas mentioned in the appendix? Other test-time approaches to ARC-AGI (e.g. Akyurek et al. 2024’s Test-Time-Tuning) seem to conclude that reusing weight adaptations across puzzles is counterproductive compared to having separate weights for each.

**Ethical Concerns:**

["NO or VERY MINOR ethics concerns only"]

**Final Justification:**

I recommend accept

**Limitations:**

potential negative social impact?

**Quality:**

3

**Strengths And Weaknesses:**

## Strengths
- This work introduces a novel test-time approach to low-data/few-shot learning regime.
- Includes an in-depth analysis of a target puzzle’s training run to demonstrate some measures of interpretability/transparency.
- Detailed discussion of future avenues for improving the work.

## Weaknesses
- Highly engineered pipeline for ARC-AGI. As useful as ARC-AGI is for measuring a necessary but not sufficient rapid adaptation ability we would expect from a system truly claiming AGI, targeting it directly as a downstream task with bespoke architecture/layers/representations may hurt the broad appeal of this method.
- Per-puzzle learning. This system is currently unable to carry forward anything between puzzles, and any adaptations are non-reusable. The authors acknowledge this and include it as a future direction in the appendix.

---

> ### Author Rebuttal · Authors · 2025-07-31
>
> We thank the Reviewer for their thoughful evaluation of our work.
>
> We completely agree with all the strengths and weaknesses listed.
>
> **Discussion on weakness #1:**
>
> Indeed, our system includes many components that target ARC-AGI specifically, and we are wary that parts of our contribution may not extend outside of this domain. So, we tried our best to focus on the more general, less ARC-AGI specific contributions, by frontloading these main contributions into Sections 1 and 3, and pushing the less transferrable contributions back towards the Appendix. For example, our conversion of MDL into a training algorithm is not specific to ARC-AGI since it can be applied to any kind of data infilling problem. We would appreciate any advice on how to better demarcate the parts of our work that are less targeted at ARC-AGI, that readers can take away and apply to other problems.
>
> **A note about weakness #2:**
>
> The restriction that our model does not carry information forward between puzzles can conversely be seen as a challenge to tests our system's generalization ability, amplifying the significance of this aspect. Having this inter-puzzle sharing restriction can be seen as either positive or negative depending on how you want to look at it.
>
> **Answering Questions:**
>
> 1. Yes, you are understanding it correctly, the neural network (specifically storing the weights) is highly load bearing, and the differentiable loss that we use is an underestimate. That said, if we presume the weights take up a large constant load (e.g., by storing the weights fully as float32 values), then including/excluding the load of the neural network weights makes no difference to the final inference procedure. This is because the constant added load goes away when differentiating the loss for the training. For MDL purposes, it is not ideal to assume we store the full weights as float32, and if we could efficiently compress them instead as a modification of our current system, this would drastically improve our results. In this direction, there is a possibility to store them in the same format as how we are compressing the neural network inputs, but we expected such a strategy to cause difficulties with posterior collapse in the training dynamics, and did not try this.
>
> 1. This work is mostly a proof of concept for the MDL approach towards intelligence. Most of the methodology is not transferrable to other downstream tasks. The main part that is transferrable, is the idea of how to transform the MDL objective into a problem of gradient descent during inference time, as explained in Section 1. This is because the statements of Section 3 still hold when we replace:
>     - "solving ARC-AGI puzzles" -> "infilling partially observed data"
>     - "incomplete puzzle" -> "observed data"
>     - "answers" -> "unobserved data"
>
>     Looking at our method as a data infilling methodology via the above replacements may help other researchers make similar derivations of further methods for other domains like supervised learning and generative modeling.
>
> 1. We understand that other studies conclude that model differentiation per puzzle is preferred, and you would like to see any evidence we may have against this to support our ideas for improvement. That said, we do not have early results available for the effects of sharing information between multiple puzzles. In place of this, perhaps we can provide an argument on why we think joint puzzle solving is still a good direction for our work to proceed. Namely, studies which uphold the benefits of model differentiation generally do so from the reference point of having completely shared weights for all puzzles, whereas our system is currently on the other extreme, with completely separate weights for all puzzles. Our best guess is that the best degree of sharing is somewhere in between.
>
> We thank the Reviewer once again for the discussion, and we hope we have addressed their questions satisfactorily. We invite the Reviewer to discuss further on any aspect of our work, if they so desire.

---

### Decision · Program_Chairs · 2025-09-17

**Decision:**

Reject

**Comment:**

The paper introduces CompressARC, a test-time learning method that minimizes description length to solve ARC-AGI puzzles without any pretraining. Without using the pre-provided training set, CompressARC still solves a diverse set of ARC-AGI puzzles, achieving around 20% success on the evaluation set, demonstrating that intelligence can emerge from compression rather than massive pretraining.

The idea is novel, and the results look good. However, reviewers raised concerns about clarity, as many details were missing or placed only in the appendix. During the author-reviewer discussion, the authors made efforts to address these concerns, adding explanations and new baselines. However, in the final discussion, although most reviewers were positive, none strongly advocated for acceptance. After carefully considering the entire discussion, I believe the current form does not meet the high standards of a prestigious venue such as NeurIPS due to its presentation issues, and I recommend rejecting this paper for this time. I encourage the authors to further improve the work and resubmit it to another venue.